# Snow processes in mountain forests: Interception modeling for coarse-scale applications

Nora Helbig[1], David Moeser[2], Michaela Teich[3,4], Laure Vincent[5], Yves Lejeune[5], Jean-Emmanuel Sicart[6], and Jean-Matthieu Monnet[7]

[1]WSL Institute for Snow and Avalanche Research SLF, Davos, Switzerland
[2]USGS, New Mexico Water Science Center, Albuquerque, U.S.
[3]Austrian Research Centre for Forests (BFW), Innsbruck, Austria
[4]Department of Wildland Resources, Utah State University, Logan, UT, U.S.
[5]Univ. Grenoble Alpes, Université de Toulouse, Météo-France, CNRS, CNRM, Centre d'Études de la Neige, Grenoble, France
[6]Univ. Grenoble Alpes, CNRS, IRD, Grenoble INP, Institut des Géosciences de l'Environnement (IGE) - UMR 5001, F-38000 Grenoble, France
[7]Univ. Grenoble Alpes, INRAE, LESSEM, F-38402 St-Martin-d'Hères, France

**Correspondence:** Nora Helbig, (norahelbig@gmail.com)

**Abstract.** Snow interception by forest canopy controls spatial heterogeneity of subcanopy snow accumulation leading to significant differences between forested and non-forested areas at a variety of scales. Snow intercepted by forest canopy can also drastically change the surface albedo. As such, accurately modeling snow interception is of importance for various model applications such as hydrological, weather and climate predictions. Due to difficulties in direct measurements of snow interception, previous empirical snow interception models were developed at just the point scale. The lack of spatially extensive data sets has hindered validation of snow interception models in different snow climates, forest types and at various spatial scales and has reduced accurate representation of snow interception in coarse-scale models. We present two novel empirical models for the spatial mean and one for the standard deviation of snow interception derived from an extensive snow interception data set collected in an evergreen coniferous forest in the Swiss Alps. Besides open site snowfall, subgrid model input parameters include the standard deviation of the DSM (digital surface model) and/or the sky view factor, both of which can be easily pre-computed. Validation of both models was performed with snow interception data sets acquired in geographically different locations under disparate weather conditions. Snow interception data sets from the Rocky Mountains, U.S., and the French Alps compared well to modeled snow interception with a Normalized Root-Mean-Square Error (NRMSE) for the spatial mean of $\leq$ 10 % for both models and NRMSE of the standard deviation of $\leq$ 13 %. Compared to a previous model for spatial mean interception of snow water equivalent the presented models show improved model performances. Our results indicate that the proposed snow interception models can be applied in coarse land surface model grid cells provided that a sufficiently fine-scale DSM is available to derive subgrid forest parameters.

# 1 Introduction

Snow interception is the amount of snow captured in a forest canopy. As much as 60 % of the cumulative snowfall may be
retained in evergreen coniferous forests (Pomeroy and Schmidt, 1993; Pomeroy et al., 1998; Storck and Lettenmaier, 2002).
In deciduous forests in the southern Andes as much as 24 % of total annual snowfall may be retained (Huerta et al., 2019).
Due to the sublimation of intercepted snow, a large portion of this snow never reaches the ground (Essery et al., 2003) and
the interplay of interception and sublimation creates significant below-forest heterogeneity in snow accumulation. Rutter et al.
(2009) estimated that 20 % of the seasonal snow cover in the Northern Hemisphere is located within forested areas. As such,
the mass balance of solid precipitation in forested regions, characterized by strong spatial variability of snow accumulation,
is a large contributor to the global water budget. Accurately modeling the spatial distribution of snow in forested regions is
thus necessary for climate and water resource modeling over a variety of scales (see Essery et al., 2009; Rutter et al., 2009).
Furthermore, intercepted snow can drastically change land surface albedo values in forested regions. Previous studies observed
large albedo differences (a range of 30 %) between snow-free and snow-covered forest stands (e.g. Roesch et al., 2001; Bartlett
and Verseghy, 2015; Webster and Jonas, 2018). Thus, in mountainous areas where forested and alpine regions coexist, accurate
estimates of forest albedo play a key role in correctly modeling the surface energy balance. Due to the connectivity between
interception and albedo, formulations of surface albedo over forested areas necessitate estimates of intercepted snow (e.g.
Roesch et al., 2001; Roesch and Roeckner, 2006; Essery, 2013; Bartlett and Verseghy, 2015).

To date, direct snow interception measurements have only been retrieved from weighing trees. These measurements are
limited to the point scale, are resource intensive sampling and only allow for analysis of small to medium size trees, or tree
elements (Schmidt and Gluns, 1991; Hedstrom and Pomeroy, 1998; Bründl et al., 1999; Storck and Lettenmaier, 2002; Knowles
et al., 2006; Suzuki and Nakai, 2008). However, there are indirect techniques that allow for estimations of interception over
larger spatial scales. Indirect measurements that compare snow accumulation between open and forest sites allow for a larger
spatial sampling, but may be affected by other forest snow processes, such as unloading of the intercepted snow. As such,
sample timing of snow storm conditions needs to be evaluated (e.g. Satterlund and Haupt, 1967; Schmidt and Gluns, 1991;
Hedstrom and Pomeroy, 1998; Moeser et al., 2015b; Vincent et al., 2018). Until recently, snow interception could not be
characterized over length scales on the order of several tens of meters. However, at these scales snow interception can spatially
vary due to canopy heterogeneity. The extensive data set of indirect snow interception measurements in evergreen coniferous
forests (further referred to as coniferous forest) in eastern Switzerland collected by Moeser et al. (2015b) is likely the first data
set that allows a thorough spatial analysis of snow interception.

Several statistical models for forest interception of snow water equivalent ($I_{SWE}$) have been suggested using a variety
of canopy metrics and functional dependencies for the rate and amount of storm snowfall (e.g. Satterlund and Haupt, 1967;
Schmidt and Gluns, 1991; Hedstrom and Pomeroy, 1998; Hellström, 2000; Lundberg et al., 2004; Andreadis et al., 2009;
Moeser et al., 2015b; Huerta et al., 2019; Roth and Nolin, 2019). Though these models have been demonstrated to perform
well, they often rely on detailed forest canopy density and structure metrics that are either not readily available or cannot easily

be upscaled, limiting functionality in models where the mean of model grid cells over several hundreds of meters to a few kilometers is required, which potentially reduces validity in large scale modeling efforts.

Traditional forest metrics used to parameterize snow interception include leaf area index ($LAI$), canopy closure ($CC$) and canopy gap fraction ($GF$) or sky view. These are mainly derived from hemispheric photographs ($HP$) taken from the forest floor looking upwards. However, these indices can also be estimated from synthetic hemispheric photographs ($SP$). $SP$ images mimic $HP$ images but are generated from aerial LiDAR (light detection and ranging) data. This requires the inversion of LiDAR to a ground perspective and conversion from a Cartesian to a polar coordinate system (Moeser et al., 2014). Prior work has also used return density ratios of LiDAR, which is computationally faster but less accurate than $SP$ images (Morsdorf et al., 2006). Canopy structure, or the position of a canopy element relative to the surrounding forest canopy, has also been used to model snow interception. However, as pointed out by Moeser et al. (2015b), some forest structure metrics such as $LAI$ and $CC$ are highly cross-correlated. Therefore, Moeser et al. (2015b, 2016) expanded on prior interception models (which mostly rely on the highly cross-correlated traditional forest density parameters $LAI$ and $CC$) by introducing uncorrelated, novel forest structure metrics. Their empirical interception model utilizes total open area, mean distance to canopy and $CC$. While the latter parameter was derived from $SP$ (Moeser et al., 2014), the first two parameters were directly computed from a digital surface model (DSM). Total open area is defined as the total open area in the canopy around a point, and mean distance to canopy defines how far away the edge of the canopy is from a point. Recently Roth and Nolin (2019) extended mean distance to canopy vertically, by deriving it for 1 m horizontal slices that were normalized with the corresponding elevation above the ground.

Due to the difficulties in measuring snow interception, previous empirical snow interception models were not validated in different snow climates, forest types or at varying spatial scales. During SNOWMIP2 (Essery et al., 2009; Rutter et al., 2009) 33 snow models were validated at individual forested as well as open sites, and many models used the snow interception parameterization from Hedstrom and Pomeroy (1998). This interception model was one of the first that used canopy metrics ($LAI$ and $CC$), although a snow interception model for larger scales also requires the greater canopy structure. Overall, SNOWMIP2 showed that maximum snow accumulation predictions had large errors compared to observed values in most models, but snow cover duration was well estimated. Furthermore, a universal best model could not be found because model performances at forest sites varied. This may explain why there is still no common ground with several snow-related variables in land surface models (Dirmeyer et al., 2006), which led to the current Earth System Model-Snow Model Intercomparison Project (ESM-SNOWMIP) showing overall larger errors in simulated snow depth on forest sites than on open sites (Krinner et al., 2018). Recently Huerta et al. (2019) validated three snow interception models developed for coniferous forests with observed point snow interception values in a deciduous southern beech- (*Nothofagus-*) forest of the southern Andes. All three empirical models required recalibration, with the recalibrated Hedstrom and Pomeroy (1998) model showing the overall best performance. Similarly, model simulations of Vincent et al. (2018) largely overestimated observed accumulated snow depth in a spruce forest at Col de Porte in the southeastern French Alps. They attribute this to errors in the processes linked to the snow interception model based on Hedstrom and Pomeroy (1998) due to an underestimation of the melt of intercepted snow. In a maritime climate previous snow interception models also failed to accurately model snow interception (Roth and Nolin, 2019).

While Roth and Nolin (2019) successfully modeled snow interception in a maritime climate, their model consistently underestimated snow interception in a continental climate forest. Overall, this demonstrates the need for more robust parameterizations of the processes affecting snow under forest, which is an important challenge for global snow modeling.

When modeling at resolutions greater than the point scale, accurate implementation of forest snow processes necessitates not just the mean of a grid cell but the standard deviation within a grid cell or model domain. However, to our knowledge, the standard deviation of snow interception has not yet been quantified. In this paper, we propose empirical parameterizations for the spatial mean and standard deviation of snow depth interception ($I_{HS}$ and $\sigma_{I_{HS}}$) derived from indirect interception measurements at sites with length scales on the order of several tens of meters. We analyzed an extensive data set consisting of several thousand interception measurements collected immediately after storm events in a discontinuous coniferous forest stand in the eastern Swiss Alps (Moeser et al., 2014, 2015a, b). From a LiDAR DSM with elevations $z$ (Moeser et al., 2014), we derived two canopy structure metrics: (1) the standard deviation of the DSM ($\sigma_z$) in order to represent the spatial heterogeneity of surface height in a forested model domain and (2) spatial mean sky view factor ($F_{\text{sky}}$), which roughly represents the spatial mean canopy openness but is derived here on the DSM from geometric quantities that describe the received radiative flux fraction emitted by another visible surface patch (i.e. canopy patches) (Helbig et al., 2009). These two metrics were correlated to spatial means and standard deviation of the indirect interception measurements. We validated the novel models with new indirect snow interception measurements from one site located in the Rocky Mountains of northern Utah, U.S. and from one site located at Col de Porte in the southeastern French Alps.

## 2    Data

In this study we only used indirect snow depth interception measurements. Indirect snow interception data was obtained from comparing new snow depth accumulation on the ground between open and forest sites. As such, snow depth interception (further referred to as snow interception) leads to reduced snow depth on the ground at forest sites. This indirect measurement technique allows for a collection of snow interception data over a larger area and also to investigate spatial snow interception variability. We used three snow interception data sets: One from the eastern Swiss Alps for the development of snow interception models, and two data sets for the independent validation of the developed snow interception models. One from the Rocky Mountains of northern Utah in the U.S. and one from the southeastern French Alps. In each of the three data sets snow interception was derived slightly different which is described in the following.

### 2.1    Eastern Swiss Alps

Indirect interception measurements were collected in seven discontinuous coniferous forest stands near Davos, Switzerland at elevations between 1511 m and 1900 m above sea level (a.s.l.) consisting of primarily Norway spruce (*Picea abies*) (Fig. 1a). Mean annual air temperature in Davos (1594 m a.s.l.) is approximately 3.5 °C and the average solid precipitation is 469 cm per year (climate normal 1981-2010, https://www.meteoswiss.admin.ch). The field sites are maintained and operated by the Snow Hydrology group of the WSL Institute for Snow and Avalanche research SLF in Davos, Switzerland. The sites were chosen

to limit influence of slope and topographic shading while capturing as much diversity as possible in elevation, canopy density and canopy structure (see canopy height models (CHM) of two field sites in Fig. 2). All seven field sites were equipped in the same manner and consisted of 276 marked and georectified measurement points (about $\pm 50$ cm) over a 250 m$^2$ surface area (yellow inlet in Fig. 1a corresponds to each yellow dot). Two non-forested reference sites (open field sites) (see blue dots in Fig. 1a) were equipped with 50 measurements points each to derive the average open site snowfall (accumulated snowfall).

During the winters of 2012/2013 and 2013/2014, snow depth was measured immediately after every storm with greater than 15 cm depth of snowfall in the open site. In total, nine storm events met the following pre-storm and storm conditions that allowed for indirect interception measurements: (1) no snow in canopy prior to a storm event, (2) defined crust on the underlying snow, and (3) minimal wind redistribution during the storm cycle. New snow was measured down to the prior snow layer crust from the top of the newly fallen snow layer to represent total snow interception. Total snowfall was measured at the open field sites. Snow interception was obtained by subtracting the total snowfall measured in the forest from the total snowfall measured at the open field site. The extensive measurement data set used in this study is described in high detail in Moeser et al. (2014, 2015a, b). Pre-processing resulted in 13'994 usable individual measurements from which 60 site based mean and standard deviation values of snow interception were computed. These 60 values were then utilized to develop the interception parameterizations. For all individual measurements, a mean snow interception efficiency (interception / new snowfall open) of 42 % was measured with values ranging from 0 to 100 %. The probability distribution function ($pdf$) of all snow interception data can be fitted with a normal distribution (positive part) with a Root-Mean-Square Error (RMSE) of the quantiles between both distributions of 0.6 cm and a Pearson correlation $r$ of 0.99 for the quantiles (Fig. 3). Average storm values of air temperatures covered cold (-12.1 °C) to mild (-1.9 °C) conditions.

A 1-m resolution gridded LiDAR DSM was generated from a flyover in the summer of 2010 and encompasses all eastern Swiss Alps field sites (see Fig. 1a for the extent). The initial point cloud had an average density of 36 points/m$^2$ (all returns) and a shot density of 19 points/m$^2$ (last returns only). The 1-m resolution LiDAR DSM is used for the derivation of the canopy structure metrics, the standard deviation of the DSM ($\sigma_z$) and the spatial mean sky view factor ($F_{sky}$) over each 50x50m$^2$ field site.

## 2.2 Rocky Mountains of northern Utah, U.S.

For the first validation data set, indirect interception measurements were collected at Utah State University's T.W. Daniel Experimental Forest (TWDEF; 41.86°N, 111.50°W), which is located at $\sim$2700 m a.s.l. in the Rocky Mountains of northern Utah (Fig. 1b). The forest stand is predominantly coniferous and is composed of Engelman spruce (*Picea engelmannii*) and subalpine fir (*Abies lasiocarpa*). However, deciduous quaking aspen (*Populus tremuloides*) forest stands are also present. Mean annual air temperature is approximately 4°C and mean annual precipitation is approximately 1'080 mm (PRISM Climate Group, 2012). On average 80 % of the precipitation falls as snow. Similar to the sites in the eastern Swiss Alps, two forested sites and one non-forested site were chosen limiting influences of slope and topographic shading while capturing diversity in canopy density and canopy structure. At both forested sites, measurements were taken along 20-m forested transects every 0.5 m before and after storm events. The after storm event transect was parallel to the before storm event transect but displaced by 0.5 m to avoid

impacts from the before storm event transect (yellow inlet in Fig. 1b corresponds to each yellow dot). At one non-forested reference site (open field site) (see blue dots in Fig. 1b) several disordered measurements were conducted within a fenced meadow site (20x20 $m^2$) (see blue dot in Fig. 1b). Additionally, an automatic weather station nearby provided continuous mea-
surements (Usu Doc Daniel SNOTEL site) (purple dot in Figure 1b). Because the purpose of the Utah measurement campaigns was not to measure snow interception but rather to investigate spatial variability of snow characteristics below different forest canopies (Teich et al., 2019), the derivation of snow interception differed slightly from the Swiss sites. Accumulated snowfall was first estimated as the difference between pre- and post-storm total snow depth. Then snow interception was calculated by subtracting the total snowfall derived in the forest from the total snowfall derived at the open field site.

During winter 2015/2016 several measurement campaigns took place. We selected those campaigns that allowed to reliably derive snow interception from total snow depth measurements before and after storm events. At one of the forested sites we used four parallel 20-m transects (i.e. two storm events) and at a second forested site two parallel 20-m transects (i.e. one storm event). Every time total snow depth was also measured at the non-forested meadow location (open site). Post-storm measurements were made between approximately 1 to 3 days after a recent snowfall but the total time period between every first and second campaign lasted several days including multiple snowfalls. The storm events were also temporally close, so that trees may not have been snow free prior to new snowfall. As such, unloading and snow settling may have influenced these measurements. After parsing the data to further reduce such influences, 95 individual interception measurements remained, resulting in three site based mean and standard deviation values. For all individual measurements, a mean snow interception efficiency of 33 % was measured with values ranging from 2 to 93 %. The *pdf* of all individual snow interception data can be similarly well fitted with a normal distribution (positive part) with a RMSE of the quantiles between both distributions of 1.3 cm and a Pearson correlation $r$ of 0.98 for the quantiles (Fig. 3). Average storm values of air temperatures covered cold (-7.3 °C) to mild (-1.4 °C) conditions.

A 1-m resolution gridded LiDAR DSM was generated from a flyover in July of 2009 and encompasses all field sites (Mahat and Tarboton, 2012; Teich and Tarboton, 2016) (see Fig. 1b for the extent). The initial point cloud had on average 7 returns/$m^2$ and 5 last returns/$m^2$ (shot density). The 1-m resolution LiDAR DSM is used for the derivation of the canopy structure metrics $\sigma_z$ and $F_{sky}$ over each 20-m transect (field site).

### 2.3 Southeastern French Alps

For the second validation data set, indirect interception measurements were collected in a coniferous forest stand next to the mid-altitude experimental site Col de Porte (45.30°N, 5.77°E) at 1325 m a.s.l. in the Chartreuse mountain range in the French Alps (more site details in Morin et al. (2012); Lejeune et al. (2019)). The forest stand is dominated by Norway spruce (*Picea abies*), with young silver fir (*Abies alba*) in the understory. Small deciduous trees are present along the northwest border of the experimental site. Mean annual air temperature is 6°C and the average solid precipitation at Col de Porte is 644 mm per year. All snow depth measurements were taken by the Snow Research Center (Centre d'Etude de la Neige (CEN)) in Grenoble, France as part of the Labex SNOUF project (SNow Under Forest) (Vincent et al., 2018) (Fig. 1c). There were three 8-m transects, each consisting of eight 1-m x 0.39-m wooden boxes that were aligned along the north, south and west axes of the

field site. New snow depth was measured inside each box after a storm event, and the box was then cleared of snow. Open site new snow depth measurements were obtained from snow board measurements at the experimental site. The boards were cleaned after each precipitation event. Interception was then derived as the difference between the open site and under-canopy new snow box measurements.

During winter 2017/2018 several measurement campaigns were conducted. Four snow storm events were selected after which new snow depth was measured in all boxes. Snow depth was collected after a major storm event took place. Unloading was visually observed from webcams and had a minimal influence on the measurements. A total of 96 individual interception measurements (4x24 measurements) resulted in four site based mean and standard deviation values. For the individual measurements, a mean snow interception efficiency of 66 % was measured with values ranging from 1 to 94 %. The *pdf* of all snow interception data can be roughly fitted with a normal distribution (positive part) with a RMSE of the quantiles between both distributions of 1.1 cm and a Pearson correlation $r$ of 0.96 for the quantiles (Fig. 3). Average storm values of air temperatures covered mild (-0.9 °C) to warm (1.7 °C) conditions.

A 1-m resolution gridded LiDAR DSM was generated from flyovers between 30 August and 2 September 2016 encompassing the entire Col de Porte experimental site (IRSTEA, Grenoble (see Fig. 1c)). The initial LiDAR point cloud had an average density of 24 points /m$^2$ and a shot density of 17 points/ m$^2$ (last return). The initial point cloud right at the transects had an average density of 42 points /m$^2$ and a shot density of 25 points/ m$^2$ (last return). The 1-m resolution LiDAR DSM is used for the derivation of the canopy structure metrics $\sigma_z$ and $F_{sky}$ over the three 8-m transects.

## 3 Methods

Subgrid parameterizations were derived for site means and standard deviations of snow interception using forest structure metrics and open site snowfall. We parameterize mean and spatial variability of snow interception for a model grid cell by accounting for the unresolved underlying forest structure (subgrid parameterization). Forest structure metrics are derived from DSM's to integrate both the terrain elevation and vegetation height.

### 3.1 Forest structure metrics

The sky view factor $F_{sky}$ describes the proportion of a radiative flux received by an inclined surface patch from the visible part of the sky to that obtained from an unobstructed hemisphere (Helbig et al., 2009). $F_{sky}$ is a commonly applied model parameter when computing surface radiation balances and can be easily computed for large areas from DSM's. $F_{sky}$ integrates previously applied forest structure metrics, such as total open area and mean distance to canopy, because this parameter is able to account for distance, size and orientation of individual surface (or canopy) patches (Helbig et al., 2009). We therefore selected $F_{sky}$ to parameterize the site mean and standard deviation of snow interception ($I_{HS}$, $\sigma_{HS}$). Here, we compute $F_{sky}$ from view factors which are geometrically derived quantities. They can be computed by numerical methods described within the radiosity approach for the shortwave (SW) radiation balance over complex topography (Helbig et al., 2009) and were originally introduced to describe the radiant energy exchange between surfaces in thermal engineering (Siegel and Howell,

1978). Thereby, Helbig et al. (2009) solve the double area integral using uniform but adaptive area subdivision for surface patches $A_I$, $A_J$. $F_{sky}$ for each surface patch $A_I$ is one minus the sum over all $N$ view factors $F_{IJ}$ by assuming the sky as one large surface patch. $F_{sky}$ is computed for each fine-scale grid cell of the DSM:

$$F_{I,sky} = 1 - \sum_{J=1}^{N} F_{IJ} = 1 - \sum_{J=1}^{N} \frac{1}{A_I} \int\limits_{A_I} \int\limits_{A_J} \frac{\cos\vartheta_I \,\cos\vartheta_J}{\pi \, r_{IJ}^2} \,\mathrm{d}A_I \,\mathrm{d}A_J \,. \tag{1}$$

Deriving $F_{sky}$ via Eq. (1) can account for holes in the surface, i.e. small gaps between leaves and branches in forest canopy, provided the DSM is of a high enough resolution to capture this. In this study, the employed DSM's did not resolve small gaps between branches. Common methods to derive $F_{sky}$ for forested regions is from sine and cosine weighted proportions of sky pixels of $HP$ or $SP$ as suggested e.g. by Essery et al. (2008) or from $LAI$ (e.g. Roesch et al., 2001). However, compared to computing $F_{sky}$ on DSM's these methods rely on extensive field work.

The main advantage in deriving $F_{sky}$ on DSM's is that $F_{sky}$ can be derived spatially by averaging all fine-scale $F_{sky}$ within a coarse grid cell. Here, we use the spatial mean of the sky view factor $F_{sky}$ Eq. (1) over a field site which is comparable to the spatial mean canopy openness.

The second forest structure metric selected was the standard deviation of the DSM $\sigma_z$ of a field site. Though not totally uncorrelated from the spatial mean $F_{sky}$ (Pearson $r$=-0.48), we selected $\sigma_z$ to serve in coarse-scale models that are not able to rely on computational expensive pre-computations of $F_{sky}$ on fine scales, such as land surface models covering regions of several hundreds to thousands of kilometers. $\sigma_z$ is thought to represent the spatial variability of canopy height and terrain elevation of the field site (or model domain).

## 3.2 Subgrid parameterization for forest canopy interception

Modeling the impact of forest canopy on snow accumulation on the ground involves several processes such as interception, unloading, melt and drip, and sublimation. Here, we present novel models for the spatial mean and standard deviation of snow interception. Modeling not only the mean but the standard deviation of snow interception within a grid cell or model domain opens new possibilities to describe the spatially varying snow cover in large grid cells. Empirical parameterizations for site mean and standard deviation of snow interception are derived from the 60 measured mean and standard deviation values from the Swiss data set. Estimates derived using the new models were validated from a comparison to the mean and standard deviation values from the French and U.S. field sites.

Snow interception $I$ was modeled as snow depth $HS$, i.e. $I_{HS}$, and not as snow water equivalent $SWE$, i.e. $I_{SWE}$. Snow interception models for $SWE$ would be advantageous for model applications because this removes uncertainties of the consequent empirical snow density parameterization in each model application. However, at the moment similar spatial $SWE$ interception measurements comparable to the extensive, spatial snow depth interception data set from Switzerland are not available. The reason similar $SWE$ data sets do not exist is probably that $SWE$ measurements require much more effort and are more time-consuming. We further refrained from deriving a spatial $SWE$ data set from the spatial $HS$ interception data set to avoid any potential error introduced when empirically converting measured $HS$ values to $SWE$. Thus, any future snow

density model developments should not affect our snow interception models. Previous interception models (Hedstrom and Pomeroy, 1998; Moeser et al., 2015b; Roth and Nolin, 2019; Huerta et al., 2019, e.g.) estimated new snow density to convert $HS$ into $SWE$. Models of new snow density typically rely on average storm temperature. Thus, converting $HS$ empirically to $SWE$ and then developing an empirical interception model introduces additional uncertainty. Prior work has shown a standard error of 9.31 kg/m$^{-3}$ when using estimates of density (Hedstrom and Pomeroy, 1998). As such, the snow interception
parameterizations developed here are for $HS$.

From here on, all references will be to site values (mean and standard deviation) without explicitly mentioning the 'mean', unless otherwise stated.

### 3.3 Performance measures

We use a variety of measures to validate the parameterizations: the RMSE, Normalized Root-Mean-Square Error (NRMSE,
normalized by the range of measured data (max-min)), Mean-Absolute Error (MAE), the Mean Absolute Percentage Error (MAPE, absolute bias with measured-parameterized normalized with measurements), the Mean Percentage Error (MPE, bias with measured-parameterized normalized with measurements) and the Pearson correlation coefficient $r$ as a measure for correlation. Finally, we evaluate the performance of our parameterizations by analyzing the $pdf$'s. We use the two-sample Kolmogorov-Smirnov test (K-S test) statistic values $D$ (Yakir, 2013) for the $pdf$'s (nonparametric method) and compute the
NRMSE for Quantile-Quantile plots (NRMSE$_{\text{quant}}$, normalized by the range of measured quantiles (max-min))) for probabilities with values in $[0.1, 0.9]$.

## 4 Results

### 4.1 Grid cell mean snow interception

#### 4.1.1 Model for grid cell mean intercepted snow depth

We parameterized grid cell mean intercepted snow depth ($I_{HS}$) by scaling open site accumulated snowfall $P_{HS}$ using the forest structure metrics $F_{\text{sky}}$ and $\sigma_z$. From these three variables, the interception measurements of the development data set correlated best with $P_{HS}$ ($r = 0.70$). Snow interception efficiency ($I_{HS}/P_{HS}$) correlations were slightly stronger for $\sigma_z$ ($r = 0.71$) than for $F_{\text{sky}}$ ($r = $ -0.69).

While it is clear that accumulated snowfall is the key parameter for modelling snow interception by forest canopy and as such
regulates its magnitude, the shape of the interception curve is predominantly controlled by forest canopy parameters and the interception model form itself. This behaviour of the interception curve has been recently demonstrated by comparing various $SWE$ interception models at single forest sites (Roth and Nolin, 2019). To decide on the interception model form we considered previously commonly applied functional relationships with accumulated snowfall such as from Hedstrom and Pomeroy (1998) and Moeser et al. (2015b) as well as simple relationships such as a power law. Together with our observed correlations
of the forest structure metrics $F_{\text{sky}}$ and $\sigma_z$ with snow interception efficiency we developed two statistical parameterizations for

$I_{HS}$ using two different base functions to scale $P_{HS}$ with either $F_{sky}$ and $\sigma_z$ (Eq. (2)) or with only $\sigma_z$ (Eq. (3)):

$$I_{HS} = P_{HS}^{a}\, b\, \frac{(1 - F_{sky})^{c}\, \sigma_z^{c}}{1 + exp(-d(P_{HS} - f))} \tag{2}$$

with constant parameters: a $= 0.09$ ($\pm 1.08$), b $= 0.19$ ($\pm 0.79$), c $= 0.72$ ($\pm 0.11$), d $= 0.13$ ($\pm 0.04$) and f $= 16.44$ ($\pm 16.33$) and

$$I_{HS} = P_{HS}^{a^\star}\, b^\star\, \sigma_z^{c^\star} \tag{3}$$

with constant parameters: $a^\star = 0.82$ ($\pm 0.12$), $b^\star = 0.0035$ ($\pm 0.0036$) and $c^\star = 0.80$ ($\pm 0.14$). The constant parameters result from fitting non-linear regression models by robust M-estimators using iterated reweighed least squares (see R v3.2.3 statistical programming language robustbase v0.92-5 package (Rousseeuw et al., 2015)). The 90 % confidence intervals of the parameters are given in parentheses. In both equations $P_{HS}$ and $\sigma_z$ are in cm.

The accuracy of a derived model between $I_{HS}$ and $P_{HS}$ depended upon the forest structure metrics and the underlying function applied in the potential models. While we investigated previously suggested functional dependencies for the amount of storm snowfall the best performances were seen when the base function between $I_{HS}$ and $P_{HS}$ was either a power law or a combination of a power law with an exponential dependence. Similar base functions were obtained for fine-scale $I_{SWE}$ models by Moeser et al. (2015b) (exponential) and recently by Roth and Nolin (2019) (power law).

Estimated $I_{HS}$-values from Eq. (2) or (3) increase with increasing $P_{HS}$, increasing $\sigma_z$ or decreasing $F_{sky}$. This implies that with increasing forest density (i.e. larger $\sigma_z$), $I_{HS}$ increases faster with increasing $P_{HS}$. Note that here, a lower $F_{sky}$ value denotes more pronounced forest gaps since it is derived from aerial LiDAR DSM.

Eq. (2) and (3) differ in two ways. First, Eq. (2) incorporates the functional dependency for increasing $P_{HS}$ that snow interception efficiency (interception/snowfall) increases with increasing precipitation due to snow bridging between branches until a maximum is reached after which it decreases due to bending of branches under the load (sigmoid curve as suggested by Satterlund and Haupt (1967); Moeser et al. (2015b)). Additionally, a power law dependency for accumulated open site storm snowfall is applied to force the sigmoid distribution to zero at very small snowfall events. The sigmoid curve alone is not able to reach zero, potentially breaking the mass balance. In contrast, Eq. (3) solely employs the power law dependency between $I_{HS}$ and accumulated open site storm snowfall $P_{HS}$. The second difference between both equations is that Eq. (2) uses both forest structure metrics ($F_{sky}$ and $\sigma_z$), whereas Eq. (3) only uses $\sigma_z$. Eq. (2) is thus more 'complex', and necessitates more time to derive both forest structure parameters whereas Eq. (3) has a more 'compact' form and solely necessitates estimation of $\sigma_z$.

To evaluate model performances with respect to a simple baseline interception estimate we linearly fitted the key parameter accumulated snowfall to measured snow interception by assuming constant impact of forest canopy parameters, i.e. $I_{HS} = cc\ P_{HS}$ with constant fit parameter cc $= 0.40$ ($\pm 0.03$).

### 4.1.2 Validation of model for grid cell mean intercepted snow depth

Performances of both newly developed snow interception $I_{HS}$ models (Eq. (2) and (3)) were compared to the $I_{HS}$ measurements from the development data set (Switzerland), as well as the $I_{HS}$ measurements from the combined validation data sets

(France and U.S.). In Figs. 4 to 6 we differentiate the validation data set from the development data set by using a black outline around the symbols (validation) instead of colored circles (development). Squares represent the data set from the U.S. and diamonds represent the data set from France.

Fig. 4 shows that for both models, there is a good agreement for $I_{HS}$ to measured interception at all sites. Overall error statistics show good performances for the development and the validation data sets with low absolute errors (e.g. all MAE$\leq$1.2 cm), strong correlations (all $r \geq$0.89) and low distribution errors (e.g. all NRMSE$_{\text{quant}}$<8 %) (Table 1). In contrast to the validation data sets performance statistics for the development data set are slightly reduced for the more compact model (Eq. (3)) compared to the more complex model (Eq. (2)). Overall, the performance metrics suggest that the simple baseline interception model is worse for both the development and the validation data sets (I(c) in Table 1).

Fig. 5 reveals overall similar performances for both parameterizations as a function of accumulated new snowfall. However, small differences between both parameterizations are visible in the extremes, i.e. for very low and very large $I_{HS}$ and $P_{HS}$. The bias for the largest $P_{HS}$ (U.S. data set) is larger for the more compact parameterization (Eq. (3)) whereas for the smallest $P_{HS}$ (data set from France) the bias is slightly larger for the more complex parameterization (Eq. (2)). The bias is more pronounced with regard to the corresponding interception efficiencies, shown in Fig. 5d-f, the largest bias for the smallest $P_{HS}$ for the complex parameterization (Eq. (2)) is -0.24 compared to 0.21 for the more compact parameterization (Eq. (3)).

## 4.2 Grid cell standard deviation of snow interception

### 4.2.1 Model for standard deviation of snow depth interception

We parameterized the standard deviation of snow depth interception $\sigma_{I_{HS}}$ by scaling $P_{HS}$ using the forest structure metric $\sigma_z$. $\sigma_{I_{HS}}$ of the development data set correlated best with $P_{HS}$ ($r = 0.82$). The correlation with mean snow interception $I_{HS}$ was less pronounced ($r = 0.33$). $\sigma_{I_{HS}}$ normalized with $P_{HS}$ correlated much better with $\sigma_z$ ($r = $ -0.68) than with $F_{\text{sky}}$ ($r = 0.13$).

Building upon the observed power law functional dependency between mean snow interception $I_{HS}$ and $P_{HS}$ and the observed relationships and correlations for $\sigma_{I_{HS}}$ we scaled a power law function for $P_{HS}$ with the standard deviation of the DSM $\sigma_z$ in order to parameterize $\sigma_{I_{HS}}$:

$$\sigma_{I_{HS}} = P_{HS}^{\text{g}} \frac{\text{h}}{1 + \sigma_z^{\text{j}}} \, . \tag{4}$$

Constant parameters g= 0.78 ($\pm$*0.10*), h= 13.40 ($\pm$*11.64*) and j= 0.53 ($\pm$*0.12*) result from fitting a non-linear regression model, similar to the derivation of $I_{HS}$ from Eq. (2) and (3). The 90 % confidence intervals of the parameters are given in parentheses. In Eq. (4) $P_{HS}$ and $\sigma_z$ are in cm.

$\sigma_{I_{HS}}$ derived from Eq. (4) increases with increasing $P_{HS}$ or decreasing $\sigma_z$. This implies that with decreasing $\sigma_z$ (decreasing forest density), the spatial variability in snow interception increases faster with increasing $P_{HS}$. The opposite correlation was found between $\sigma_z$ and mean snow interception $I_{HS}$. For a $\sigma_z$ converging to zero, modeled $\sigma_{I_{HS}}$ via Eq. (4) approaches a constant fraction of precipitation.

Similarly to the derivation of a baseline estimate for our $I_{HS}$ models, we linearly fitted accumulated snowfall to measured standard deviation of snow interception to evaluate the model performance with respect to a simple baseline estimate for the standard deviation of snow interception. This resulted in $\sigma_{I_{HS}} = \text{jj } P_{HS}$ with constant fit parameter jj $= 0.20$ ($\pm 0.01$).

### 4.2.2 Validation of model for standard deviation of snow depth interception

Overall, modeled and measured $\sigma_{I_{HS}}$ agree well (Fig. 6). Error statistics show good performances for the development and the validation data set with low absolute errors (e.g. all MAE$\leq$0.63 cm), strong correlations (all $r \geq$0.92) and low distribution errors (e.g. NRMSE$_{\text{quant}}$<10 %) (Table 1). However, performances are less accurate for the validation data set than for the development data set (e.g. MAE of 0.63 cm as opposed to 0.45 cm and NRMSE$_{\text{quant}}$ of 10 % as opposed to 4 %). This was caused by a potential outlier in the validation data set from the U.S. During one measurement campaign, an open site

accumulated storm snowfall $P_{HS}$ was not available at the same date as the under canopy measurements. Therefore, this value was estimated from a local automatic weather station (Usu Doc Daniel SNOTEL site; purple dot in Figure 1b). Additional measurement uncertainty (at the Utah site) was also introduced, since interception estimates were integrated values over several snow storms that occurred during the 13 days between pre- and post- snowfall measurement campaigns. When this outlier is removed from the validation data set, performance statistics improve considerably converging towards the errors of the

development data set, cf. MAE decreases to 0.35 cm and the NRMSE$_{\text{quant}}$ to 5 %.

Overall, the performance of the baseline model for $\sigma_{I_{HS}}$ is worse compared to our model performance (II(b) in Table 1). However, because one observed $\sigma_{I_{HS}}$ of the validation data set from the U.S. (2.9 cm) was better estimated by the baseline model compared to our model (4 cm compared to 5.2 cm), the NRMSE and RMSE for the baseline estimates were somewhat better.

To compare modeled (Eq. (2) and Eq. (4)) and measured data set mean values from each geographic location (Switzerland, U.S., France), we averaged all site values to derive an overall mean of $I_{HS}$, and $\sigma_{I_{HS}}$ for each location. The coefficient of variation (description of variability) ($CV_{I_{HS}}$=$\sigma_{I_{HS}}/I_{HS}$) was also calculated for each of the three geographic locations. For the Swiss development data set, the same overall mean, standard deviation and $CV$ for measured and modeled snow interception was calculated (mean of 9.4 cm, standard deviation of 4.5 cm and $CV$ of 0.51). For the validation data sets we

obtained slightly larger values for modeled $I_{HS}$ (9.3 cm), modeled $\sigma_{I_{HS}}$ (3.7 cm) and modeled $CV_{I_{HS}}$ (0.38) than measured $I_{HS}$ (9.2 cm), measured $\sigma_{I_{HS}}$ (3.2 cm) and measured $CV_{I_{HS}}$ (0.35). If the potential outlying data point from Utah is removed, the same overall modeled and measured mean $CV_{I_{HS}}$ (0.32) is found along with very close values of modeled and measured mean $I_{HS}$ (9.8 cm versus 9.9 cm) and modeled and measured $\sigma_{I_{HS}}$ values (3.4 cm versus 3.3 cm).

### 5 Discussion

We proposed two empirical models for spatial mean interception $I_{HS}$ to be employed in hydrological, climate and weather applications. One model is a more compact model, Eq. (3). This model uses a power law dependency between $I_{HS}$ and accumulated storm precipitation $P_{HS}$ that is scaled by one forest structure metric: the standard deviation of the DSM $\sigma_z$. The

other model, Eq. (2), integrates a more complex parameterization by using a combination of a power law with an exponential dependence similar to the one suggested by Moeser et al. (2015b) for $P_{HS}$ and is scaled by two forest metrics: the sky view factor $F_{\mathrm{sky}}$ in combination with $\sigma_z$. For both $I_{HS}$ models, interception increases faster with increasing snowfall when forest density increases (i.e. larger $\sigma_z$). In the more complex model increasing forest density is implemented by increasing $\sigma_z$ and decreasing $F_{\mathrm{sky}}$. Though $F_{\mathrm{sky}}$ can be pre-computed and is temporally valid for many years (unless the forest structure changes due to logging, fires, insect infestations or other forest disturbances), computing $F_{\mathrm{sky}}$ over large scales and/or with fine resolutions is more computationally demanding than for $\sigma_z$ (Helbig et al., 2009). A subgrid parameterization for the sky view factor of coarse-scale DSM's over forest canopy would eliminate the pre-computation of sky view factors on fine-scale DSM's. Such a subgrid parameterization for sky view factors over forest canopy could be similarly set up as previously done for alpine topography and would lead us towards a global map of sky view factors (cf. Helbig and Löwe, 2014).

In general, more differences between the compact and more complex modeling approaches only displayed at the extremes. For instance, for small storm precipitation values ($P_{HS}$ =3 cm), the more compact parameterization performs slightly better whereas for very large storms ($P_{HS}$ =43 cm), the more complex model displayed improved performance. The choice of one of these two models thus depends on the focus range of precipitation values and available computational resources.

Our choice for the functional form of $P_{HS}$ differs from previous parameterizations for snow interception solely using the sigmoid growth $\sim 1/(1+exp(-k(P-P_0)))$ (e.g. Satterlund and Haupt, 1967; Schmidt and Gluns, 1991; Moeser et al., 2015b) or an exponential form $\sim (1-exp(-k(P-P_0)))$ (e.g. Aston, 1993; Hedstrom and Pomeroy, 1998) with increasing precipitation. While the base function of Satterlund and Haupt (1967) worked better for Moeser et al. (2015b), a drawback of this relationship is that interception does not become exactly zero for a zero snowfall amount. To account for this, the model becomes complicated when applied to discrete model time steps (Moeser et al., 2016). For this reason, Mahat and Tarboton (2014) selected the relationship proposed by Hedstrom and Pomeroy (1998) for their parameterization of snow interception. However, the functional form of the Hedstrom and Pomeroy (1998) model does not account for snow bridging or branch bending, thus modeling interception efficiency as decreasing through time. We also compared means and standard deviations over all sites as a function of forest metrics and found that the use of storm means can introduce precipitation dependencies that might originate from an insufficient number of sites showing similar forest canopy structure parameter values for a given precipitation (cf. black line compared to colored dots in Fig. (5)). Based on the functional dependencies revealed by analyzing our data as a function of $P_{HS}$ and forest structure metrics, a simple power law was able to describe the spatial mean $P_{HS}$ dependency of snow interception (cf. Eq. (3)). The equation displayed that with increasing $P_{HS}$, $I_{HS}$ increases. This is less pronounced with smaller $\sigma_z$ or larger $F_{\mathrm{sky}}$ values (Fig. (5)). Very recently, a storm event power law dependency was also found to best describe fine-scale $SWE$ interception in a maritime snow climate (Roth and Nolin, 2019). Our base functions for site means and standard deviations thus bear some similarity to previously developed fine-scale snow interception models. Despite an ongoing debate regarding the proper representation of interception, we believe that the interception models presented here have the advantage that they could be applied in various model applications for horizontal grid cell resolutions larger than a few tens of meters. Due to the lack of measurements over larger scales a validation remains at the moment impossible.

We derived just one empirical model for the standard deviation of snow interception $\sigma_{I_{HS}}$ that uses a power law dependency on accumulated storm precipitation $P_{HS}$ scaled by one forest structure metric: the standard deviation of the DSM $\sigma_z$. We also tested a more complex model for $\sigma_{I_{HS}}$ using both forest metrics ($F_{\text{sky}}$ and $\sigma_z$) that integrates a power law dependency of $P_{HS}$. However, model performances for the validation data set did not differ considerably from the ones for the more compact model. Therefore, we propose the more compact parameterization for $\sigma_{I_{HS}}$ (Eq. (4)) to facilitate broad model applications.

By using $F_{\text{sky}}$ and $\sigma_z$ derived from DSM's as forest structure metrics we focused on the overall shape of the forest. This simplification is similar to the assumption by Sicart et al. (2004) for solar transmissivity in forests under cloudless sky conditions. They assumed the fraction of solar radiation blocked by the canopy was equal to 1-$V_f$ with $V_f$ being defined as the fraction of the sky visible from beneath the canopy. Our simplification is also in line with previous suggestions. Primarily, to reliably describe interception by forest canopy over larger areas, the larger-scale canopy structure needs to be taken into account instead of only using point based canopy structure parameters (e.g. Varhola et al., 2010; Moeser et al., 2016). We proposed to calculate $F_{\text{sky}}$ and $\sigma_z$ on DSM's rather than on CHM's to account for terrain and vegetation height. This results from our correlation analysis for measurement data collected in rather flat field forest sites (Section 2) and should be verified once spatial snow interception measurements become available in steeper terrain and over larger length scales.

The models for $I_{HS}$ and $\sigma_{I_{HS}}$ were statistically derived from measured snow interception data gathered in the eastern Swiss Alps. Naturally, empirically derived parameterizations can only describe data variability covered by the data set. However, even though the parameterizations were developed empirically, we could display that the parameterizations perform well for two disparate, independent snow interception data sets collected in geographically different regions, different snow climates, coniferous tree species and prevailing weather conditions during collection of the validation data sets (French Alps and Rocky Mountains, U.S.). For instance, in the French Alps, rather warm to mild winter weather conditions predominated whereas rather mild to cold weather prevailed during the campaigns in the Rocky Mountains of northern Utah in the U.S. Though snow cohesion and adhesion are clearly temperature dependent, we did not observe decreases in overall performances under these differing weather conditions for our two $I_{HS}$ models, which do not include air temperature. In contrast, in a maritime (warm) snow climate correlations between air temperature and snow interception were recently found by Roth and Nolin (2019). In addition to the spread in observed temperature conditions, our ranges of accumulated snow storm $P_{HS}$ values of the development data set are fairly broad (e.g. $P_{HS}$ between 10 cm and 40 cm). The measurements of the validation data set are well within the range of the development data set values, but also cover extremes, such as one very small ($P_{HS}$= 3 cm) and one very large snowfall ($P_{HS}$= 43 cm) (cf. Fig. 3). It is thus reassuring that our models perform sufficiently well in varying climate regions; however more validation data sets would be advantageous especially in regions experiencing extreme climates such as the cold arctic or warm maritime forests. Despite the existing variability in the data set, more spatial snow interception measurements would clearly help to increase the robustness of our empirical parameterizations.

To date, interception models have been created for $SWE$ instead of snow depth and were mostly point models instead of spatial mean interception parameterizations. As such, a comparative assessment (beyond the independent validation sets in the body of this paper) of our models to absolute performance measures of previous interception models was difficult. However, we calculated relative error estimates for an inter-model comparison with two $SWE$ interception models. We selected the

empirical, recently developed $SWE$ model from Roth and Nolin (2019) as well as the 50x50 m$^2$ stratified $SWE$ model (for 50x50 m$^2$ grid cell size) from Moeser et al. (2015b). The Moeser et al. (2015b) model utilized the same Swiss data set as this study, which is currently the largest set of spatial interception measurements in the world. The Roth and Nolin (2019) model error estimates were calculated for a subset of their data set which included three snowfall events and interception values acquired at three elevations under mild temperature conditions in the McKenzie River Basin Oregon, U.S. (for details on data see Table 2 in Roth and Nolin, 2019). We estimated a NRMSE of 28.9 %, a MPE of -5.7 % and a MAPE of 31.2 % for the three modelled and measured interception values. The Moeser et al. (2015b) model error estimates were calculated for a subset of the Swiss data set consisting of 34 spatial mean observed interception values (50x50 m$^2$) and 34 parameterized values. We estimated a NRMSE of 9.3 %, a MPE of -16.5 % and a MAPE of 23.5 %. Compared to previous $SWE$ interception models, we obtain improved performances (using means of error estimates over I(a) and I(b) respectively in Table 1). The fairest comparison is the one with the stratified $SWE$ model of Moeser et al. (2015b) compared to which our mean error estimates show a 9 % respectively a 4 % reduction in the NRMSE, a 60 % respectively a 75 % reduction in the MPE and a 40 % respectively a 50 % reduction in the MAPE for the more complex model (Eq. (2)) respectively more compact model (Eq. (3)). Our improved model performances as compared to prior interception models in tandem with a good performance for two distinctly different validation data sets lend validity to improving coarse-scale climate and hydrologic (watershed and snow) model applications.

Despite the overall good performance of the models, we observed differences between the two validation data sets. The data set collected in France shows improved error statistics for snow interception $I_{HS}$ (e.g. for Eq. (3): RMSE=0.35 cm, NRMSE=4 %, MAE=0.26 cm) as compared to the data set collected in the U.S. (e.g. for Eq. (3): RMSE=1.52 cm, NRMSE=14 %, MAE=1.4 cm). In France, intercepted snow storm depth was measured as the difference of new snow depth in wooden boxes below trees and open site new snow storm depth. This was done in relatively short time intervals after a snow storm. In the U.S., intercepted snow was inferred from total snow depth before and after a snow storm event within forests and in an open site. Derived snow interception was often integrated over several storm events due to longer periods between measurement campaigns. Thus, these measurements were potentially influenced by other snow and forest processes such as snow settling, wind redistribution, sublimation, unloading, and melt and drip. Our interception models however only calculate how much snow is intercepted at any point in time, which provides the input for other forest snow process models such as for unloading, sublimation as well as melt and drip. We thus assume that these processes will be addressed separately, as in all prior interception models (Roesch et al., 2001). Despite some uncertainties in the validation data set from the U.S. it allowed for validation in a different snow climate than the French Alps and also covered a large spread in storm snowfall amounts (Fig. 4).

Differences in model performances between the two validation data sets could also be attributed to the more accurate forest structure metrics for the French data set because of a higher resolution LiDAR DSM (higher point density of 24 /m$^2$ returns and 17 /m$^2$ last returns) compared to the LiDAR flyover from the U.S. (on average 7 returns/m$^2$ and 5 last returns/m$^2$). While it is clear that the higher the point cloud density, the greater the potential detail of derived DSM's, 1-m resolution DSM's computed from point clouds above 5 returns/m$^2$ are usually quite consistent, and are suitable to derive coniferous canopy models allowing

tree-level analyses (Kaartinen et al., 2012; Eysn et al., 2015). Current available or scheduled country-wide data sets are now around 1-5 returns/m$^2$ (e.g. Federal Office of Topography Swisstopo, last access: 22 November 2019; Danish Geodata Agency, last access: 22 November 2019; Latvian Geospatial Information Agency, last access: 22 November 2019) and these densities can be expected to increase thanks to technical improvements in LiDAR sensors. Since fine-scale DSM's are the only input required to derive the forest structure metrics $F_{sky}$ and $\sigma_z$ a global applicability of our snow interception models for coniferous forest would be possible with minimal required information.

To understand if the models would also work in other forest types or in disturbed forests, e.g. due to logging, fires or insect infestations, more snow interception measurements in deciduous and mixed as well as disturbed forests are required. Very recently Huerta et al. (2019) showed that previously published snow interception models developed for coniferous forests from Hedstrom and Pomeroy (1998); Lundberg et al. (2004); Moeser et al. (2016) required recalibration to match observed point snow interception observations in a deciduous southern beech *Nothofagus* stand of the southern Andes. We investigated the performance of our models for two measurement campaigns in a deciduous quaking aspen (*Populous tremuloides*) forest in our U.S. field site. The measurement setup (20-m transects) was identical to the ones in the coniferous forest at this location (see Section 2.2). Though overall the models compared well with the measurements, the model performance was not as good as for the coniferous forest. Because the LiDAR DSM was acquired in the summer, i.e. with leaves on the trees, the models naturally overestimated $I_{HS}$ and $\sigma_{I_{HS}}$. For instance, using the more complex model for $I_{HS}$ (Eq. (2)) we obtained a mean bias of -6 cm, whereas when using the more compact model for $I_{HS}$ (Eq. (3)) we obtained a mean bias of -8 cm. For $\sigma_{I_{HS}}$, the performance was overall slightly better with a mean bias of -3 cm (Eq. (4)). While this shows that the performance is clearly lower in such sites, we assume that the performance would be improved when the LiDAR is acquired in leaf-off conditions.

The LiDAR-derived DSM sky view factors do not account for small spaces between leaves or branches, which are well accounted for when sky view factors are derived from $HP$ or $LAI$. In principle, sky view factors that are computed on DSM's represent, depending on the return signal used to create the DSM, a coarser view on the underlying forest canopy. While this increases the overall fine-scale error, we feel that the ability to calculate both our canopy structure metrics in the Cartesian DSM space, which allows an easy model application, far outweighs fine-scale resolution losses.

## 6 Conclusion and Outlook

The statistical models for spatial mean and standard deviation of snow interception presented here are a first step towards a more robust consideration of snow interception for various coarse-scale model applications. They were built upon a very large dataset and validated by two other datasets from different geographic regions and snow climates, and performed well for all three sites and under differing weather conditions. For spatial mean interception all NRMSE's were $\leq 10$ % and for the standard deviation of interception all NRMSE's were $\leq 13$ %. Compared to a previous model for spatial mean $SWE$ at 50x50 m$^2$ the presented models for spatial mean snow interception show improved model performances.

In our observed snow interception datasets, as much as 68 % and on average 43 % of the cumulative snowfall (accumulated snowfall of snowfall event in cm) was retained by coniferous forests (interception efficiency (snow interception/accumulated

snowfall) of site means) and as much as 14 % and on average 11 % was retained by deciduous forests. These values compare well to previously observed snow interception in coniferous trees reaching up to 60 % of cumulative snowfall (Pomeroy and Schmidt, 1993; Pomeroy et al., 1998; Storck and Lettenmaier, 2002) and to 24 % of total annual snowfall in a deciduous forest in the southern Andes (Huerta et al., 2019).

The empirical models integrate forest parameters that can be derived from fine-scale DSM's, which can be pre-generated and
515 stored for large regions. One of the presented interception models only relies on the standard deviation of the fine-scale DSM, which is a very efficient way to integrate snow interception in coarse-scale models such as land surface models. This could greatly improve current forest albedo estimates and the subsequent surface energy balance for various model applications such as hydrological, weather and climate predictions.

However, the presented parameterizations were developed and validated for spatial means and standard deviations over
520 horizontal length scales of a few tens of meters. We can only hypothesize that the parameterizations are also valid at coarser length scales due to the use of non-local forest structure parameters. Representative non-local forest structure parameters require that a DSM of high enough resolution is available to represent subgrid variability of forest structure in the coarse-scale model grid cell. However, there was and probably is, to date, no validation data available at large spatial scales. The investigated length scale matches current satellite resolutions (e.g. Sentinel and Landsat), which opens further cross-validation
and deployment opportunities with satellite-derived parameters such as surface albedos and fractional-snow covered area. With parameterizations for both the mean and standard deviation of snow interception by forest canopy, the distribution of intercepted snow depth in forests can now be derived whenever a sufficiently high-resolution DSM is available.

*Data availability.* The data that support the findings of this study are available from the corresponding author upon reasonable request. (norahelbig@gmail.com).

*Competing interests.* The authors declare that they have no conflict of interest.

*Acknowledgements.* We thank E. Thibert and É. Mermin, who performed GPS and tree measurements of the site setup at Col De Porte, and J.A. Lutz, S. Jones, J. Carlisle and D.G. Tarboton for their support and the possibility to work at TWDEF. N. Helbig was partly funded by the Federal Office of the Environment FOEN. M. Teich was partly funded by the Swiss National Science Foundation (P2EZP2_155606, P300P2_171236), the Utah Agricultural Experiment Station (UAES), and the USU Department of Wildland Resources. The Labex SNOUF
project has been supported by a grant from Labex OSUG2020-ANR10 LABX56. The TWDEF LiDAR data processing was supported by NSF EPSCoR cooperative agreement IIA 1208732 awarded to Utah State University, as part of the State of Utah Research Infrastructure Improvement Award. Any opinions, findings, and conclusions or recommendations expressed are those of the author(s) and do not necessarily reflect the views of the National Science Foundation.

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

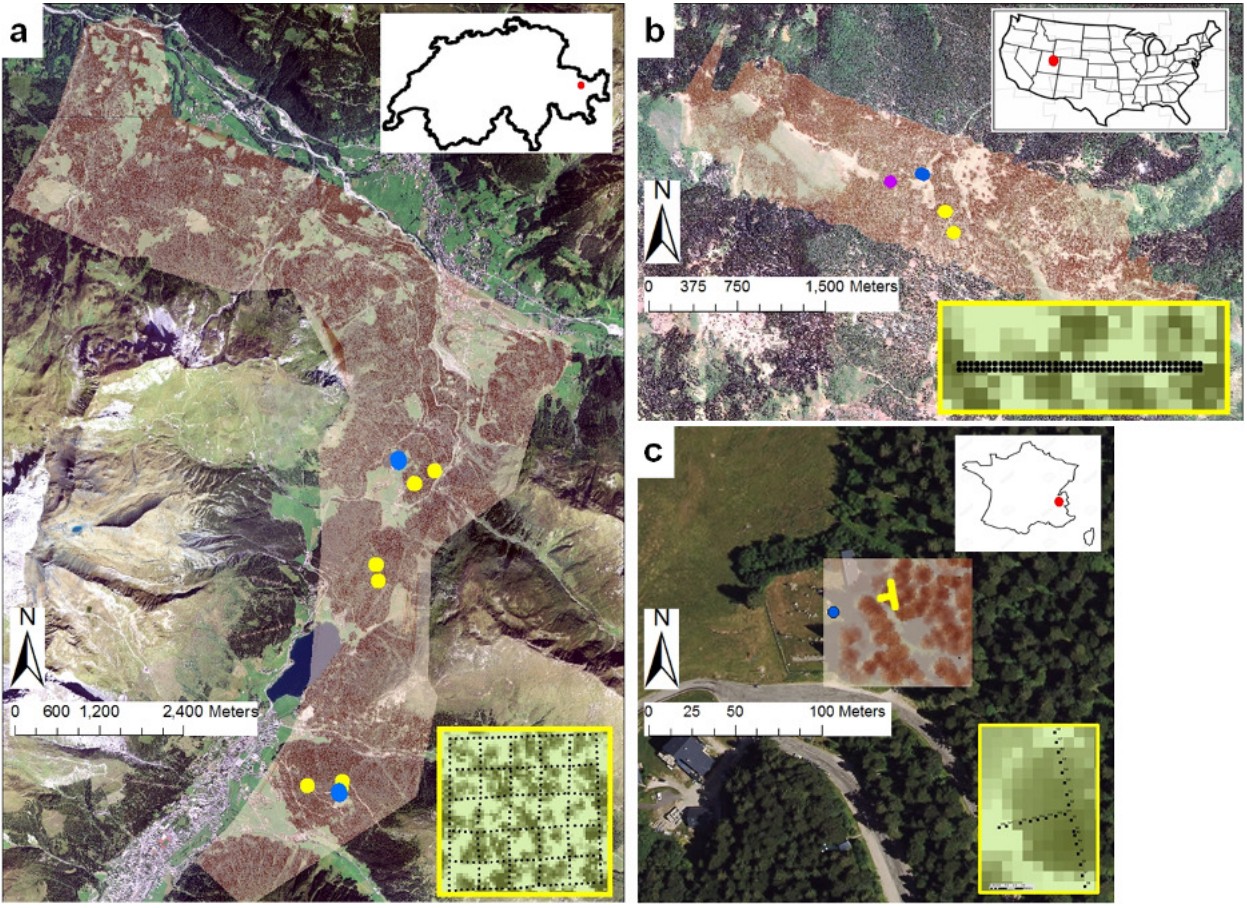

**Figure 1.** Extent of LiDAR derived canopy height model (CHM) with locations of open (blue points) and forested field sites (yellow points), and SNOTEL site (purple point): (a) close to Davos in the eastern Swiss Alps ($\sim$90 km$^2$; 46.78945°N, 9.79632°E), (b) in the Rocky Mountains of northern Utah, U.S. ($\sim$13 km$^2$; 41.85046°N, 111.52751°W), and (c) in the southeastern French Alps at Col de Porte ($\sim$0.01 km$^2$; 45.29463°N, 5.76597°E). The yellow framed inlets show the respective snow depth measurement setup at the forested field sites. Underlying orthophotos were provided for the French site by IGN (France) and for the Swiss site by Swisstopo (JA100118). For the site in the U.S. © Google Earth imagery was used.

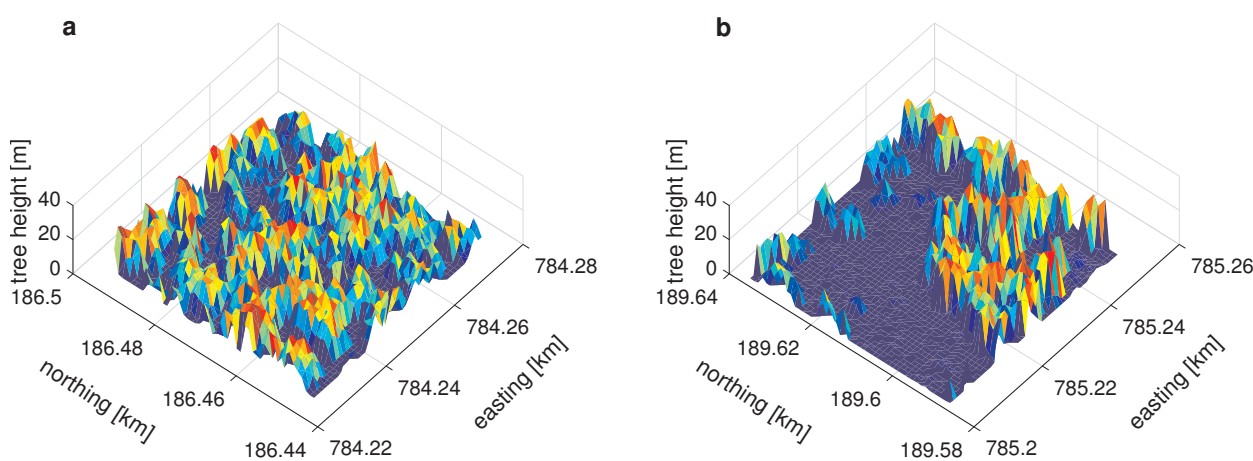

**Figure 2.** Canopy height models (CHM) for two 50 x 50 m$^2$ field sites in 1 m grid resolution in the eastern Swiss Alps with (a) high canopy coverage and (b) low canopy coverage (for detailed site descriptions see Moeser et al., 2014).

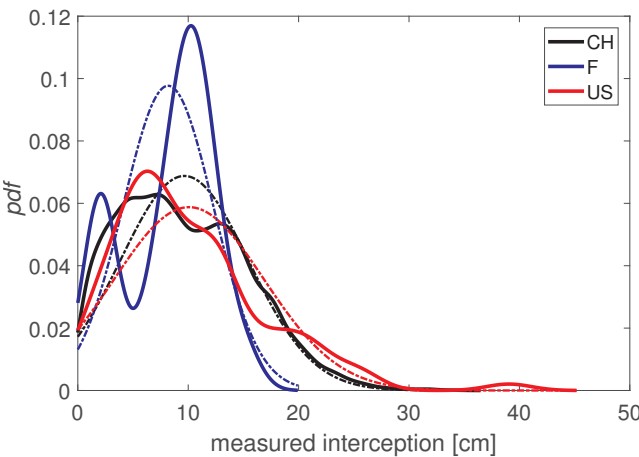

**Figure 3.** Probability density functions (*pdf*'s) of all individual snow depth interception measurements used for the development (Swiss (CH) data set) and for the validation of the parameterizations (French (F) and U.S. (US) data sets). The dashed lines indicate a theoretical normal *pdf* for the corresponding data set.

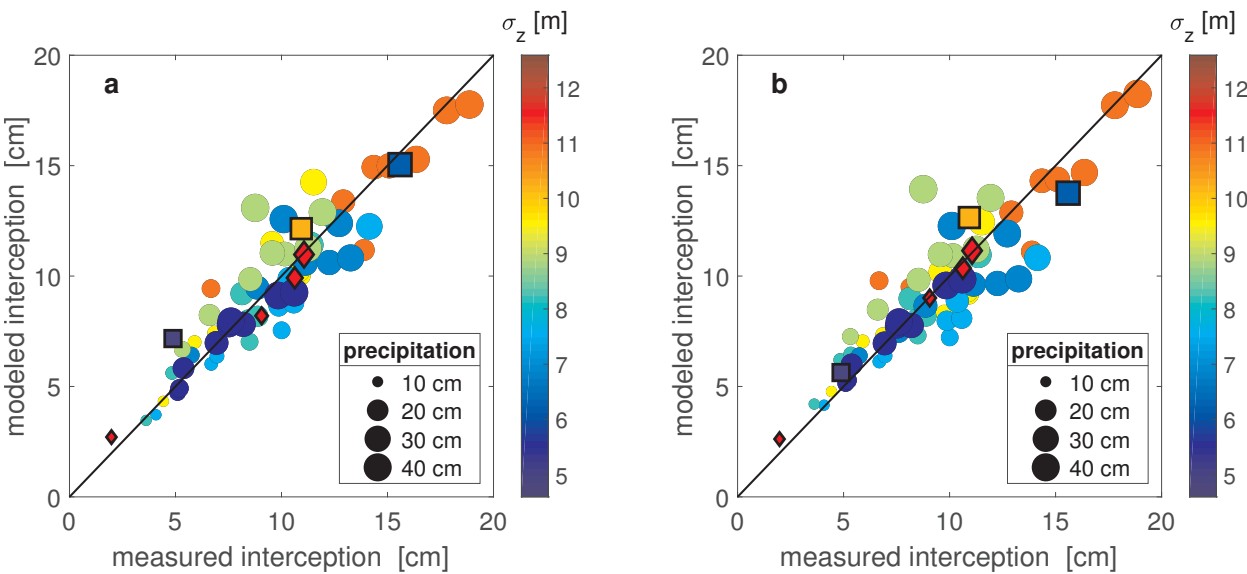

**Figure 4.** Measured and parameterized site means of intercepted snow depth, i.e. spatially averaged over each site and for each storm date. Parameterized using a) Eq. (2) and b) Eq. (3) as a function of site means of standard deviation of the LiDAR DSM $\sigma_z$ (in color) as well as open site snow storm precipitation (size of symbols). Circles represent the development data set from Switzerland, symbols with a black border represent the validation data sets with squares for that from the U.S. and diamonds for that from France.

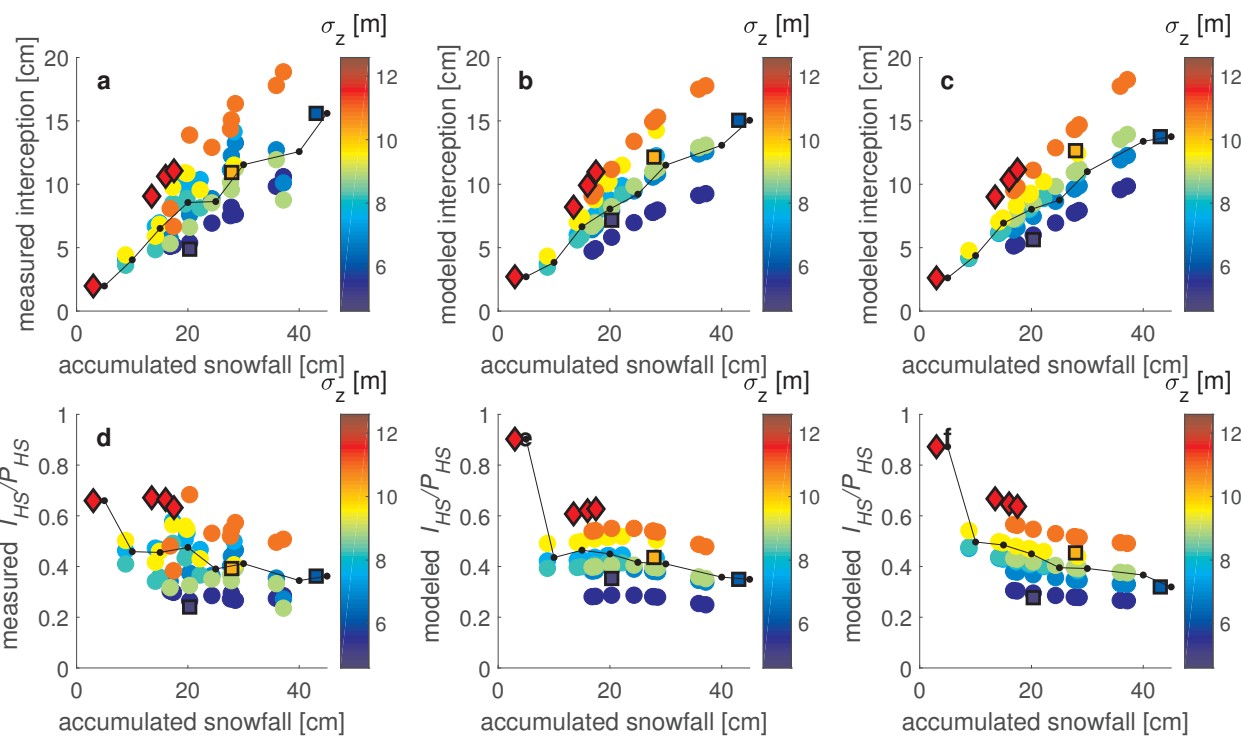

**Figure 5.** Snow depth interception $I_{HS}$ (a,b,c) and interception efficiency $I_{HS}/P_{HS}$ (d,e,f) as a function of accumulated open site snow storm precipitation $P_{HS}$ and standard deviation of the LiDAR DSM $\sigma_z$ (in color). The y-axis of the first column shows measured data, the second column shows model output with Eq. (2) and the third model output with Eq. (3). Site means for each storm event are depicted with colored circles for the development data set from Switzerland and symbols with a black border depict the validation data sets, with squares for that from the U.S. and diamonds for that from France. Storm means (in $P_{HS}$ bins) are shown in black.

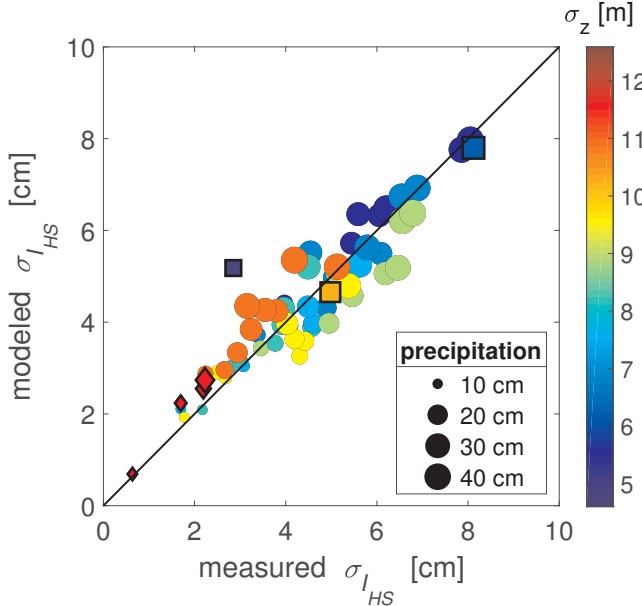

**Figure 6.** Measured and parameterized standard deviation of snow depth interception $\sigma_{I_{HS}}$ at each site and for each storm date. Parameterized using Eq. (4) as a function of site means of standard deviation of the LiDAR DSM $\sigma_z$ (in color) as well as open site snow storm precipitation (size of symbols). Circles represent the development data set from Switzerland, symbols with a black border represent the validation data sets with squares for that from the U.S. and diamonds for that from France.

**Table 1.** Performance measures between measurement and parameterization of (I) spatial-mean snow depth interception $I_{HS}$ with (a) Eq. (2), (b) Eq. (3), and (c) a baseline model and of (II) standard deviation of snow depth interception $\sigma_{I_{HS}}$ with (a) Eq. (4) and (b) a baseline model. Statistics are shown for the development data set from the eastern Swiss Alps (CH) and for the combined validation data set (U.S.&F).

| | NRMSE | RMSE | MPE | MAPE | MAE | $r$ | K-S | NRMSE$_{\text{quant}}$ |
|---|---|---|---|---|---|---|---|---|
| | [%] | [cm] | [%] | [%] | [cm] | | | [%] |
| **I $I_{HS}$** | | | | | | | | |
| **(a) Eq. (2)** | | | | | | | | |
| CH | 8.7 | 1.33 | -1.97 | 11.29 | 1.01 | 0.92 | $8.6\,10^{-2}$ | 2.5 |
| U.S.&F | 8.2 | 1.12 | -10.61 | 16.46 | 0.92 | 0.97 | $1.4\,10^{-1}$ | 7.8 |
| **(b) Eq. (3)** | | | | | | | | |
| CH | 10.2 | 1.55 | -1.65 | 12.83 | 1.15 | 0.89 | $1.0\,10^{-1}$ | 5.3 |
| U.S.&F | 7.5 | 1.03 | -7.03 | 11.28 | 0.76 | 0.97 | $2.9\,10^{-1}$ | 5.9 |
| **(c) $I_{HS} = 0.40\,P_{HS}$** | | | | | | | | |
| CH | 16.6 | 2.53 | -2.58 | 21.46 | 2.02 | 0.70 | $1.2\,10^{-1}$ | 4.2 |
| U.S.&F | 21.8 | 2.97 | 10.89 | 33.64 | 2.55 | 0.97 | $4.3\,10^{-1}$ | 16.9 |
| **II $\sigma_{I_{HS}}$** | | | | | | | | |
| **(a) Eq. (4)** | | | | | | | | |
| CH | 8.9 | 0.57 | -2.05 | 10.9 | 0.45 | 0.92 | $8.6\,10^{-2}$ | 3.9 |
| U.S.&F | 12.7 | 0.95 | -21.52 | 24.51 | 0.63 | 0.94 | $4.3\,10^{-1}$ | 10.4 |
| **(b) $\sigma_{I_{HS}} = 0.20\,P_{HS}$** | | | | | | | | |
| CH | 14.0 | 0.89 | -3.42 | 15.79 | 0.66 | 0.82 | $1.2\,10^{-1}$ | 6.3 |
| U.S.&F | 11.0 | 0.83 | -28.07 | 30.31 | 0.72 | 0.98 | $4.3\,10^{-1}$ | 12.7 |