# Peer review of "Snow processes in mountain forests: Interception modeling for coarse-scale applications"

_Hydrology and Earth System Sciences, 2019_

## Referee Comment (RC1) · Anonymous Referee #1 · 23 Sep 2019

Review of: Snow processes in mountain forests: Interception modeling for coarse-scale applications by Nora Helbig et al.

This paper presents a development and validation of a simple snow interception model that can be applied in large scale modelling efforts. Snow interception can be a significant part of cold-region forest water balances and novel approaches are needed to more robustly estimate this term especially in large scale modelling activities. Overall the paper articulates the challenge well and provides a simple solution. I would recommend publication with some minor revisions.

Major comments:

Throughout the paper forest structure is parameterized from a Digital Surface Model

(DSM). If I am reading this correctly this is a DSM that is normalized to terrain to give a DSM in terms of vegetation height? (Z-axiz of Figure 2)? To be consistent with literature this should rather be termed a canopy height model (CHM) rather than a DSM. https://www.earthdatascience.org/courses/earth-analytics-python/lidar-raster-data/lidar-chm-dem-dsm/

Can you clarify the type of model you are presenting in context of the existing models? There was much discussion of other existing models and a clear compare/contrast of what you are presenting would be beneficial. This is simple empirical relationship with 1 or two input variables rather than a physical/mechanistic parameterization. This is critical to clarify so that future users can determine how to use this moving forward.

A clearer description of what was being measured at the various sites is needed. It was not immediately apparent that all of this was based on snow depth difference only and ignored density. The assumption that density of new snow accumulation is the same between open and forested areas is critical. Is it reasonable to assume that the standard error of 9.31 kg/m2 of new snow estimates is greater than observed snow density differences between open and forested? Even immediately after snowfall events there will be differences in density associated with unloading/compaction on the dripline of tree crowns versus the influence of blowing snow redistribution/erosion or not in clearings? At these locations is it reasonable to assume snowfall is the same between forest and clearing locations – any preferential deposition patterns evident? Variable blowing snow deposition/erosion in clearings versus forests? In the end do you have any observations that you could demonstrated that density differences are negligible or provide these values in terms of SWE? Any errors in density differences could lead to relatively large errors in interception ratios, especially for small events, and this needs to be clarified.

The transferability of this model is tested by applying the Swiss parametrization to French and US sites. While results are promising for between these sites I would temper some of the speculation (339-353). Relative to the large range of climatic conditions of cold-regions forests globally these sites represent relatively warm locations. As expressed elsewhere there has been variability in interception model performances between maritime and continental locations not to mention more temperature cold regions versus cold arctic treeline/tundra locations. Before recommending this for universal and widespread applications this model should be tested if possible at other locations that represent more end members.

The approach implemented is to parameterize an empirical relationship. This will not work perfectly for all scenarios/locations obviously. Is it possible to quantify the uncertainty of the parameters and how they may vary between sites / vegetation types? How stable are these parameters?

Specific comments:

First sentence on abstract and line 19-21 are a little contradictory.

26-33: Transition to discussing surface albedo is abrupt. While snow interception/albedo is a critical feedback it is not extensively discussed hereafter? Can this section be simplified?

41-42: Awkward sentence

102: define more clearly what indirect interception measurements are.

151-154: what may the influence of different point cloud densities be upon the CHM derivation? Are there any recommendations you could make on what should be collected in future for proper a parameterization of your models?

205: how are you getting from SP a ta point to mean sky view factor. How is the space being discretized? Are you computing SP on a fine scale grid and averaging values over a coarser scale of interest?

248-250: can you clarify this reversed response?

279: Why do we want to know the standard deviation of snow interception. Can you

articulate a broader reason to calculate this?

286-287: As canopy gets to be more homogeneous the spatial variability of interception increases? How?

322-324: Could a global product (between 51.6° N and 51.6° S at least)for these metrics be derived from the GEDI platform? https://gedi.umd.edu/

365: Typo "ASifferences"

381-383: Deciduous will have very different behavior than coniferous vegetation. Could you reoptimise your model for deciduous specific sites? Would be interesting to know if the same scaling laws were applicable to know if a separate deciduous scaling parameterization is needed or not.

386-388. Why? Can you justify this a bit more?

412-416: The full summary of the various interception efficiencies would be better presented in the results rather than in the conclusion for the first time.

423-425: long challenging sentence

Figure 1: Can scale bars and north arrows be consistently sized and located on edge of orthophotos? Where snow depth measurement setups the same for each point in the respective sites?

Figure 2: what is grid resolution of DSM (aka CHM)? What UTM zone is applicable for the respective easting/northing? Correct sig figs on the easting northing?

Figure 4-6: "Parametrized" or "Modelled" interception on y-axis label?

---

## Referee Comment (RC2) · Anonymous Referee #2 · 13 Oct 2019

This manuscript describes an empirical approach to quantify snow interception. This is an important topic and the authors use interesting data sets. However, I have to admit that I felt rather confused when reading the manuscript. The language is partly ambiguous, the structure is unclear, details of the field observation and 'modelling' are missing and the applicability of the empirical equations remains unclear to me.

The language needs to be improved to be more concise. Just as one example: P1L17: Would snow in another season not be intercepted? Both in this sentence and the next one I assume the authors mean that in a coniferous forest 60% may be intercepted. As it reads now, 60% of some total are intercepted in coniferous forests and 24% are intercepted in deciduous forests in the Andes, i.e. 84% are retained in total. I agree that this is a minor detail and one can guess what the authors mean, but in a scientific

paper these things should be formulated as clearly as possible.

Central parts of the methods are described first in the result section.

The field observations need to be described in more detail. I honestly do not understand what has been measured how. It also sounds as if some data were selected from a larger set, the reasons for this are not entirely clear.

The two central equations suddenly pop up in the result section. How were these two types of equations derived? Is there any physical reasoning for certain functional relationships like the exp or power function? How exact can the coefficients be determined? Uncertainty? Sensibility? Furthermore, I do not understand what the stdev of the DSM is. Variation of ground surface? But this would not have anything to do with the trees. Variation of vegetation heights? But then DSM is the wrong term.

My major concern regarding usability is the choice to express everything as snow height rather than SWE. When used as part of a larger model, I would assume one is most often interested in SWE rather than heights. Also conceptually I am not sure what the height of intercepted snow implies? Height on branches? Probably rather height as the snow would be if being on the ground? But then at which density, that of the other snow on the ground or that of the intercepted snow? Sorry, but I find this very confusing and limiting. Thus, I would prefer to see the interception etc expressed in SWE.

As the two equations are derived from data for ideal situations (no prior snow . . .) I am not sure how these should be used for the real case, where there is often a history of prior snow on the trees. It seems here one might run into the problem that a simple empirical equation is not really a model after all. For a 'model' I would expect some canopy storage accounting, which is an aspect that is missed here.

---

## Author Comment (AC2) · 20 Nov 2019

**We thank the reviewer for their encouraging and constructive comments. Their comments (in italics) are addressed below. Answers to the reviewer are given in blue.**

*Major comments:*

*The language needs to be improved to be more concise. Just as one example: P1L17: Would snow in another season not be intercepted? Both in this sentence and the next one I assume the authors mean that in a coniferous forest 60% may be intercepted. As it reads now, 60% of some total are intercepted in coniferous forests and 24% are intercepted in deciduous forests in the Andes, i.e. 84% are retained in total. I agree that this is a minor detail and one can guess what the authors mean, but in a scientific paper these things should be formulated as clearly as possible.*

The manuscript was carefully read by all co-authors and additionally by several native English speakers from the USGS. The language was therefore carefully checked before the submission. Nevertheless, to avoid any unclarities in the manuscript, we will carefully go through the manuscript once more, and make the wording more concise where appropriate.

*Central parts of the methods are described first in the result section.*

Unfortunately we do not understand which parts you mean. All methods are described in the methods section. The result section is structured as follows:

4.1. Grid cell mean snow interception
4.1.1 Parameterization
4.1.2 Validation

4.2 Grid cell standard deviation of snow interception
4.2.1 Parameterization
4.2.2 Validation

The resulting parameterizations are not part of the method, they were developed in this study using the data given in the data section, the forest structure metrics and the method both described in the methods section (3.1 and 3.2). As such the parameterizations (i.e. 4.1.1. and 4.2.1) are part of the results. The validation sections describe how modeled interception compares to observed interception (i.e. 4.1.2. and 4.2.2).

While we do not see that parts of the results should be moved to the methods we agree that the headings could be more concise and we will consider changing them.

*The field observations need to be described in more detail. I honestly do not understand what has been measured how. It also sounds as if some data were selected from a larger set, the reasons for this are not entirely clear.*

Unfortunately, we do not understand which data you think were selected from a larger set. However, we will carefully go through the data section to clarify the description of the measurement methods.

*The two central equations suddenly pop up in the result section. How were these two types of equations derived? Is there any physical reasoning for certain functional relationships like the exp or power function? How exact can the coefficients be determined? Uncertainty? Sensibility?*

In the method section
3.2 Subgrid parameterization for forest canopy interception
we describe that the empirical parameterizations are derived using the Swiss data set (Line 211 ff).
All parameterizations were empirically developed using the Swiss development data set. The existence of varying previously observed functional relationships (base functions) were mentioned in the introduction (Line 43-46) and were considered here. In the results section (Line 243-246 and 251-259) we explain the reasoning for our functional relationship. In the discussion (Line 389-405) we largely discuss our choice compared to previously published base functions.
To specify the robustness of our coefficients we will include the confidence interval of the coefficients.

*Furthermore, I do not understand what the stdev of the DSM is. Variation of ground surface? But this would not have anything to do with the trees. Variation of vegetation heights? But then DSM is the wrong term.*
Forest structure is parameterized here from the Digital Surface Model (DSM) which is the top of the surface, i.e. surface elevation + vegetation height (DTM+CHM). DSM is a standard abbreviation for the surface height (e.g. https://www.earthdatascience.org/courses/earth-analytics-python/lidar-raster-data/lidar-chm-dem-dsm/). The abbreviation DSM was introduced in line 92.

The standard deviation of DSM $\sigma_z$ describes the variation of vegetation altitude by integrating both the variability of the canopy height and of surface elevation. Using $\sigma_z$ seems more realistic because gaps and spaces between trees are influenced by local topography. The standard deviation of DSM $\sigma_z$ was introduced in the methods section as our second forest structure metric (3 Methods / 3.1 Forest structure metrics in Line 206-210).

*My major concern regarding usability is the choice to express everything as snow height rather than SWE. When used as part of a larger model, I would assume one is most often interested in SWE rather than heights. Also conceptually I am not sure what the height of intercepted snow implies? Height on branches? Probably rather height as the snow would be if being on the ground? But then at which density, that of the other snow on the ground or that of the intercepted snow? Sorry, but I find this very confusing and limiting. Thus, I would prefer to see the interception etc expressed in SWE.*
We deliberately chose to parameterize snow depth over SWE because snow depth was spatially measured on the ground and not SWE.
We could have applied an empirical new snow parameterization to derive spatial SWE based on e.g. interpolated air temperatures (as e.g. in Moeser et al., 2015b). This would however have introduced a lot of uncertainty in an interception model since this is determined by the applied empirical density parameterization, measurement errors in air temperatures as well as by the spatial interpolation of temperatures. This was discussed in Line 214-220.
We therefore decided to derive spatial snow depth interception estimates from snow depth observations within and outside of the forest. Using our interception models spatial snow depth in forested regions can be more realistically described. Converting snow depth to/from SWE with a density parameterization and its connected uncertainties is controlled by each snow module (as part of a complex model) and these uncertainties will not be linked with the presented snow interception model.

We will expand our explanation in Line 214-220.

*As the two equations are derived from data for ideal situations (no prior snow …) I am not sure how these should be used for the real case, where there is often a history of prior snow on the trees. It seems here one might run into the problem that a simple empirical equation is not really a model after all.*

You are right our parameterizations were developed on a data set that had no prior snow on tree branches when the precipitation event started. Nevertheless, both validation data sets did not have this prerequisite but still compared well to modeled interception using the novel empirically derived parameterizations. Especially the interception data set from the US often integrated snow interception over several storms due to longer time periods between data collection. Thus, the trees weren't necessarily snow free for a following snowstorm. Instead these measurements may have been influenced by snow settling, wind redistribution, sublimation, unloading and melt.

All this was discussed in the discussion (Line 359-362).

*For a 'model' I would expect some canopy storage accounting, which is an aspect that is missed here.*

We have focused on improvements of an interception model rather than multiple related processes. Modeling forest canopy involves several processes, each of which are described with separate models. This includes, unloading, melt and drip (some models), and sublimation. These models use the interception model to dictate how much snow is in the branches at any point in time. If there is still snow in the branches, then it is depleted by the unloading, melt and drip and sublimation models. Thus, canopy storage is dictated by the interplay of each individual model.

In line 361-363 we discuss that we present a model for one forest process. We will add some extra explanation. Furthermore, we will make this clear in the method section: "3.2 Subgrid parameterization for forest canopy interception".

---

## Author Response (AR1)

**We thank the editor and reviewers for the comments to improve the quality of our manuscript. We have carefully addressed the comments with point-by-point replies to the editor and reviewers (in blue) and revised our manuscript accordingly. Attached is a marked-up version of the manuscript.**

**Reply to editor comments:**

*Editor comments to the author:*
*When revision your manuscript, please pay particular attention to the following aspects:*
*- Both referees comment on the use of snow depth of SWE. This is a critical issue. It is important to realize and recognize that snow depth is not a conservative quantity, and that there is a risk for bias when comparing snow depth observations between places with different microclimate conditions. This does not mean that there is no value in making a model based on snow depth observations, but the right arguments should be used to motivate this choice. This is not to reduce "any potential error when converting measured HS values to SWE." as is stated in the manuscript. The real reason (I guess) is that SWE observations are not available because they require more effort in sampling. But that does not make them less relevant. So please be clear in motivating the choices, and acknowledge any potential limitation and the fact that all models will use SWE and not snow depth.*

You are right. We chose to parameterize snow depth over SWE because snow depth was spatially measured on the ground and not SWE. We could have applied an empirical new snow parameterization to derive spatial SWE based on e.g. interpolated air temperatures (as e.g. in Moeser et al., 2015b). This would however have introduced more uncertainty in an interception model since this is determined by the applied empirical density parameterization, measurement errors in air temperatures as well as by the spatial interpolation of temperatures. Converting snow depth to/from SWE with a density parameterization and its connected uncertainties thus remain controlled by the snow module (as part of a complex model) and these uncertainties will not be linked with the presented snow interception model.

We made the reasons why we chose snow depth over SWE for our interception models more clear (Section "Subgrid parameterization for forest canopy interception").

*- The choice for the functional relationships that are fitted to be data need to be motivated, and if possible evaluated against alternative models. There should be some objective criteria behind the choice for a particular functional form.*

All parameterizations were empirically developed using the Swiss development data set. The existence of varying previously observed functional relationships (base functions) were mentioned in the introduction (Line 44-66) and were considered here. In the results section (276-279 and Line 283-291) we explain the reasoning for our functional relationship. In the discussion (Line 363-382) we largely discuss our choice compared to previously published base functions.

To specify the robustness of our regression coefficients we now included the confidence interval of the coefficients. Clearly, more spatial snow depth interception data sets would be advantageous. However, our empirical parameterizations are based on a development data set that is currently probably the most extensive existing data set available for spatial snow depth interception. This and the overall good performance of the parameterizations for the validation data set are reassuring that the models can be applied by various model applications. This is newly mentioned in the discussion.

*- Please pay particular attention to the writing. If possible, let the manuscript be proof-read by a native speaker/colleague.*
The manuscript was carefully read by several native English speakers from the USGS. The language was thus carefully checked before submission. Nevertheless, we went carefully through the manuscript again, and made the wording and structure more concise where appropriate.

**Additional introduced changes by the authors:**

1) For the revised manuscript, we decided to use the latest version of our code for computing the sky view factors, as this includes an improved visibility algorithm. This and the change that we now use $\sigma_z$ in cm in all equations to be conform with the applied units for snow depth resulted in some changes in the regression coefficients. However, only very minor changes result in the figures and the performance table.

2) Furthermore, to avoid infinity for an extreme value of $\sigma_z$=0 in the parameterization of the standard deviation of snow interception (Eq. (4)) but keeping the functional form we introduced $\sigma_{HS}=f(1/(1+\sigma_z))$. This slight change in the functional form has almost no impact on modeled interception values and also changed performances only slightly.

The overall outcome of our work was not affected by these additional changes.

**We thank the reviewer for their encouraging and constructive comments. Their comments (in italics) are addressed below. Answers to the reviewer are given in blue.**

*Major comments:*

*Throughout the paper forest structure is parameterized from a Digital Surface Model (DSM). If I am reading this correctly this is a DSM that is normalized to terrain to give a DSM in terms of vegetation height? (Z-axiz of Figure 2)? To be consistent with literature this should rather be termed a canopy height model (CHM) rather than a DSM.* *https://www.earthdatascience.org/courses/earth-analytics-python/lidar-rasterdata/lidar-chm-dem-dsm/*

We parameterize forest structure from a Digital Surface Model (DSM) which is the top of the surface (as defined in the short course definition in https://www.earthdatascience.org/courses/earth-analytics-python/lidar-raster-data/lidar-chm-dem-dsm/), i.e. CHM+DTM.
You are right the caption of Figure 2 was unclear. We changed that and also stress in the introduction that canopy structure metrics are derived from DSM's.

*Can you clarify the type of model you are presenting in context of the existing models? There was much discussion of other existing models and a clear compare/contrast of what you are presenting would be beneficial. This is simple empirical relationship with 1 or two input variables rather than a physical/mechanistic parameterization. This is critical to clarify so that future users can determine how to use this moving forward.*
We agree that it is important to characterize the type of model. We added "empirical" when we refer to model throughout the manuscript to make this more clear: e.g. in the abstract: "We present two novel empirical models …", in the discussion: "We proposed two empirical models for.." resp. "We have derived just one empirical model for the standard deviation of snow interception.-.." and in the conclusion: "The empirical models integrate forest parameters..".

*A clearer description of what was being measured at the various sites is needed. It was not immediately apparent that all of this was based on snow depth difference only and ignored density. The assumption that density of new snow accumulation is the same between open and forested areas is critical. Is it reasonable to assume that the standard error of 9.31 kg/m2 of new snow estimates is greater than observed snow density differences between open and forested? Even immediately after snowfall events there will be differences in density associated with unloading/compaction on the dripline of tree crowns versus the influence of blowing snow redistribution/erosion or not in clearings? At these locations is it reasonable to assume snowfall is the same between forest and clearing locations – any preferential deposition patterns evident? Variable blowing snow deposition/erosion in clearings versus forests? In the end do you have any observations that you could demonstrated that density differences are negligible or provide these values in terms of SWE? Any errors in density differences could lead to relatively large errors in interception ratios, especially for small events, and this needs to be clarified.*

We agree that snow densities can be quite different between open and forested sites, especially with compaction and redistribution of snow by wind. Therefore we chose to parameterize snow depth and not snow water equivalent (SWE), which would require empirical parameterizations for the snow density itself (see Section 3.2 Subgrid parameterization for forest canopy interception). Until now we do not have available spatial, reliable SWE measurements comparable to the data set used in this study.

Note that the Swiss data was immediately acquired following a new snow fall event such that the influence of unloading, compaction and an eventual wind impact on the snowpack can be assumed reasonably small (see Moeser et al, 2015b).
In France, snow depth data was collected within a few days after the snow fall. The operational flat field site in France shows low wind speed and has therefore only very limited snow drift. Average hourly wind speed (at 10m height) is 1.2m/s over a period of 1993-2011 (Morin et al., 2012).
The open field site in the US may have been influenced by wind redistribution and compacting that might have created differences in the snowpack.

However, the data set from France and the US were only used as independent validation data sets, and the results are promising.

We improved the data description where we make clear that we used observed snow depth differences, i.e. independent measurements.

*The transferability of this model is tested by applying the Swiss parametrization to French and US sites. While results are promising for between these sites I would temper some of the speculation (339-353). Relative to the large range of climatic conditions of cold-regions forests globally these sites represent relatively warm locations. As expressed elsewhere there has been variability in interception model performances between maritime and continental locations not to mention more temperature cold regions versus cold arctic treeline/tundra locations. Before recommending this for universal and widespread applications this model should be tested if possible at other locations that represent more end members.*
While we agree that the novel models should be tested for a broad range of climatic conditions including extreme climate conditions and at various geographic sites we believe that the three sites already cover substantial variability (as shown by mean air temperatures and precipitation sums). We therefore believe that the novel models could perform sufficiently well in other climate conditions (though of course extremes have to be investigated). At the moment we do not have more snow interception data sets available that would allow an extended evaluation.

Note however, that we also point out the limitations of the models when applied for a deciduous forest. This was discussed in line 441-454.

We rephrased the section which now also more clearly mentions the limitations to soften our recommendation (395-415).

*The approach implemented is to parameterize an empirical relationship. This will not work perfectly for all scenarios/locations obviously. Is it possible to quantify the uncertainty of the parameters and how they may vary between sites / vegetation types? How stable are these parameters?*

Yes, our approach is to parameterize an empirical relationship. This model was then validated with independent data sets at different locations and for one different vegetation type, suggesting that it is also applicable in different climates/forests. Nevertheless, it is clear that more data is required to confirm this. To specify the robustness of our fitted parameters we included the confidence interval of the estimates.

*Specific comments:*
*First sentence on abstract and line 19-21 are a little contradictory.*
This was modified.

*26-33: Transition to discussing surface albedo is abrupt. While snow interception/albedo is a critical feedback it is not extensively discussed hereafter? Can this section be simplified?*
We agree and we rephrased the transition.

*41-42: Awkward sentence*
Agreed, we rephrased this sentence.

*102: define more clearly what indirect interception measurements are.*
Indirect interception measurements were introduced in line 36-38 in the introduction. We expanded the explanation in the data section as well.

*151-154: what may the influence of different point cloud densities be upon the CHM derivation? Are there any recommendations you could make on what should be collected in future for proper a parameterization of your models?*
In general the higher the point cloud, the greater the potential detail of the model. Specifically, if multi-return LiDAR is being integrated, then the higher the density of the last returns, the higher the potential detail of the DSM. This translates into a higher resolution CHM as well, since this data is subtracted from the raw data to create canopy heights from elevations.

1-m resolution DSMs computed from points clouds above 5 returns/m$^2$ are usually quite consistent, and are generally considered as suitable from automated tree detection. Local artifacts (NA or low pixels) can be expected due to heterogeneous scanning pattern on the ground, and to canopy penetration variability depending on forest type and beam intensity and divergence. But the description of the canopy at 1 m resolution is quite robust for such densities. Higher densities are probably required in the case of deciduous species with LiDAR data acquired in leaf-off conditions.
Ten years ago, country-wide acquisitions would be typically between 0.5 to 2 returns/m$^2$. Current available or scheduled country-wide datasets are now around 1-5 returns/m$^2$ (e.g., Denmark 5 returns/m$^2$, North-Rhine Westphalia in Germany 4 returns/m$^2$, Spain 1 return/m$^2$, France 2 to 5 returns/m$^2$).
We can expect that thanks to technical improvements in LiDAR sensors, the density of 5 returns/m$^2$ will be exceeded in most countrywide campaigns in the next decade. Besides, acquisitions over smaller areas (municipalities…) have usually higher densities.

*205: how are you getting from SP at a point to mean sky view factor. How is the space being discretized? Are you computing SP on a fine scale grid and averaging values over a coarser scale of interest?*

Here, we derive the sky view factor $F$sky using Eq. (1) for each fine-scale grid cell in a DSM. A spatial mean is obtained by averaging all fine-scale grid cells within a coarse-scale grid cell. The spacing thus depends on the available fine-scale grid cell resolution of the DSM. We do not use $F$sky derived from SP. We made this clear in line 211-219.

*248-250: can you clarify this reversed response?*
Deriving the sky view factor $F$sky using Eq. (1) for a fine-scale grid cell in a DSM implies a calculation on the DSM. Deriving $F$sky from HP or SP allows a view from below canopy. Since we do not use HP derived $F$sky and to avoid confusion we removed the second part of the explanation.

*279: Why do we want to know the standard deviation of snow interception. Can you articulate a broader reason to calculate this?*
While we had some explanations at the end of the introduction we added some at the beginning of Section 3.2.

*286-287: As canopy gets to be more homogeneous the spatial variability of interception increases? How?*
The larger $\sigma_z$ the more trees are in a field area (50x50m$^2$ plot), i.e. the denser the coverage:

[Figure]

Based on our data set larger $\sigma_z$ implies larger spatial mean interception, but lower spatial variability of interception. Lower $\sigma_z$ implies lower spatial mean interception, but larger spatial variability of interception:

[Figure]

[Figure]

We will rephrase the explanation in line 320-323.

To avoid infinity for an extreme value of $\sigma_z$=0 but keeping the functional form we introduced $\sigma_{HS}$=$f(1/(1+\sigma_z))$. This new form changes modeled interception values and performances only slightly. We changed this in the revised manuscript.

*322-324: Could a global product (between 51.6_ N and 51.6_ S at least) for these metrics*
*be derived from the GEDI platform? https://gedi.umd.edu/*
Thanks a lot for pointing us towards the GEDI platform. This is a very promising mission. A GEDI product exists with 25 m grid resolution for ground elevation and canopy top height (L1A-2A). We assume a product like thus could be used to compute $F$sky spatially (Eq. (1)).

However, a 25 m fine-scale DSM is much coarser than the resolutions used here for developing a snow interception model. This might need a scale-dependent investigation.

*365: Typo "ASifferences"*
Corrected.

*381-383: Deciduous will have very different behavior than coniferous vegetation. Could you reoptimise your model for deciduous specific sites? Would be interesting to know if the same scaling laws were applicable to know if a separate deciduous scaling parameterization is needed or not.*
We agree that snow interception models should be verified for different vegetation species. As discussed in line 266- 270 Huerta et al. (2019) showed very recently that current interception models developed for coniferous vegetation required recalibrating of fit parameters to be applicable in deciduous forests. However, the same scaling laws were applicable.
Our models were developed and validated for coniferous vegetation. We further validated the models with indirect interception measurements from two measurement campaigns conducted in a deciduous forest in the US. Larger biases resulted. However, we could not perform a solid validation of our models with this data set since the LiDAR point cloud was acquired during leaves-on conditions, which led to overestimations in modeled interception.
To develop empirical interception models for deciduous forests measurement campaigns and a LiDAR acquired during leaves-off conditions are required.

*386-388. Why? Can you justify this a bit more?*
Unfortunately, it is not fully clear to us what your question is.
The novel interception models presented here, use forest structure metrics which can be derived spatially on a DSM without tedious field measurements. The accuracy of the derived metrics is dictated by the resolution of the DSM. In contrast a more accurate presentation of forest structure metrics might be achieved using field measurements (e.g. HP) but then a spatial coverage is not feasible.

*412-416: The full summary of the various interception efficiencies would be better presented in the results rather than in the conclusion for the first time.*
We prefer having this summary in the conclusions. It is not really a result, but rather characterizes observed snow interception in general compared to our datasets. It further confirms previously observed annual snow interception fractions which were mentioned at the beginning of the introduction.

*423-425: long challenging sentence*
We rephrased this sentence.

*Figure 1: Can scale bars and north arrows be consistently sized and located on edge of orthophotos? Where snow depth measurement setups the same for each point in the respective sites?*
Yes, the snow depth measurement setups were the same for each point in the respective sites. We made Figure 1 more consistent.

*Figure 2: what is grid resolution of DSM (aka CHM)? What UTM zone is applicable for the respective easting/northing? Correct sig figs on the easting northing?*
The coordinates of Figure 2 are displayed in the Swiss reference system CH1903+. It is metric similar to UTM. Grid resolution of the CHM's is 1 m as for the DSM's. This information was added to the caption.

*Figure 4-6: "Parametrized" or "Modelled" interception on y-axis label*
We changed the labels to "Modeled interception".

**Reply to reviewer2 comments :**

**We thank the reviewer for their encouraging and constructive comments. Their comments (in italics) are addressed below. Answers to the reviewer are given in blue.**

*Major comments:*

*The language needs to be improved to be more concise. Just as one example: P1L17: Would snow in another season not be intercepted? Both in this sentence and the next one I assume the authors mean that in a coniferous forest 60% may be intercepted. As it reads now, 60% of some total are intercepted in coniferous forests and 24% are intercepted in deciduous forests in the Andes, i.e. 84% are retained in total. I agree that this is a minor detail and one can guess what the authors mean, but in a scientific paper these things should be formulated as clearly as possible.*

The manuscript was carefully read by all co-authors and additionally by several native English speakers from the USGS. The language was carefully checked before the submission. Nevertheless, to we carefully went through the manuscript once more, and made the wording and structure more concise where appropriate.

*Central parts of the methods are described first in the result section.*
Unfortunately we do not fully understand which parts you mean. All methods are described in the methods section. The result section is structured as follows:

4.1. Grid cell mean snow interception
4.1.1 Parameterization
4.1.2 Validation

4.2 Grid cell standard deviation of snow interception
4.2.1 Parameterization
4.2.2 Validation

The resulting parameterizations should not be part of the methods section, they were newly developed in this study using the data given in the data section, the forest structure metrics and the method that are both described in the methods section (3.1 and 3.2). As such the parameterizations (i.e. 4.1.1. and 4.2.1) are part of the results. The validation sections (within in the results) describe how modeled interception compares to observed interception for the development and the validation data sets (i.e. 4.1.2. and 4.2.2).
While we do not see that parts of the results should be moved to the methods we agree that the headings could be more concise and we changed them to make the overall structure more clear.

*The field observations need to be described in more detail. I honestly do not understand what has been measured how. It also sounds as if some data were selected from a larger set, the reasons for this are not entirely clear.*
Unfortunately, we do not understand which data you think were selected from a larger set. However, we carefully went through the data section and clarified the description of the measurement methods where necessary.

*The two central equations suddenly pop up in the result section. How were these two types of equations derived? Is there any physical reasoning for certain functional relationships like the exp or power function? How exact can the coefficients be determined? Uncertainty? Sensibility?*

In the methods section
3.2 Subgrid parameterization for forest canopy interception
we describe that the empirical parameterizations are derived using the Swiss data set.

All parameterizations were empirically developed using the Swiss development data set. The existence of varying previously observed functional relationships (base functions) were mentioned in the introduction (Line 44-50) and were considered here. In the results section (268-269, 275-279 and 283-291) we explain the reasoning for our functional relationship. In the discussion (Line 363-388) we largely discuss our choice compared to previously published base functions.

To specify the robustness of our coefficients we now included the confidence interval of the regression coefficients and clearly discuss the need for more data in the discussion.

*Furthermore, I do not understand what the stdev of the DSM is. Variation of ground surface? But this would not have anything to do with the trees. Variation of vegetation heights? But then DSM is the wrong term.*

Forest structure is parameterized here from the Digital Surface Model (DSM) which is the top of the surface, i.e. surface elevation + vegetation height (DTM+CHM). DSM is a standard abbreviation for the surface height (e.g. https://www.earthdatascience.org/courses/earth-analytics-python/lidar-raster-data/lidar-chm-dem-dsm/). The abbreviation DSM was introduced in line 92.

The standard deviation of DSM $\sigma_z$ describes the variation of vegetation altitude by integrating both the variability of the canopy height and of terrain elevation. Using $\sigma_z$ seems more realistic because gaps and spaces between trees are influenced by local topography. The standard deviation of DSM $\sigma_z$ was introduced in the methods section as our second forest structure metric (3 Methods / 3.1 Forest structure metrics last section).

We further investigated using the standard deviation of the CHM. For our data sets this didn't change the overall functional relationship. Furthermore, correlation coefficients were larger between mean snow interception and $\sigma_z$ derived from DSM than between mean snow interception and $\sigma_z$ derived from CHM. Since all data used in this study was however collected in rather flat field sites, this may have to be verified in steeper terrain. This is now discussed.

*My major concern regarding usability is the choice to express everything as snow height rather than SWE. When used as part of a larger model, I would assume one is most often interested in SWE rather than heights. Also conceptually I am not sure what the height of intercepted snow implies? Height on branches? Probably rather height as the snow would be if being on the ground? But then at which density, that of the other snow on the ground or that of the intercepted snow? Sorry, but I find this very confusing and limiting. Thus, I would prefer to see the interception etc expressed in SWE.*

We deliberately chose to parameterize snow depth over SWE because snow depth was spatially measured on the ground and not SWE.

We could have applied an empirical new snow parameterization to derive spatial SWE based on e.g. interpolated air temperatures (as e.g. in Moeser et al., 2015b). This would however have introduced a lot of uncertainty in an interception model since this is determined by the applied empirical density parameterization, measurement errors in air temperatures as well as by the spatial interpolation of temperatures. This was discussed in Section 3.2.

We therefore decided to derive spatial snow depth interception estimates from snow depth observations within and outside of the forest. Converting snow depth to/from SWE with a density parameterization and its connected uncertainties thus remain controlled by the snow module (as part of a complex model) and these uncertainties will not be linked with the presented snow interception model.

We largely expanded our explanation in Section 3.2.

*As the two equations are derived from data for ideal situations (no prior snow …) I am not sure how these should be used for the real case, where there is often a history of prior snow on the trees. It seems here one might run into the problem that a simple empirical equation is not really a model after all.*

You are right our parameterizations were developed on a data set that had no prior snow on tree branches when the precipitation event started. Nevertheless, both validation data sets did not have this prerequisite but still compared well to modeled interception using the novel empirically derived parameterizations. Especially the interception data set from the US often integrated snow interception over several storms due to longer time periods between data collection. Thus, the trees weren't necessarily snow free for a following snowstorm. Instead these measurements may have been influenced by snow settling, wind redistribution, sublimation, unloading and melt.

This was discussed in Line 424-426.

*For a 'model' I would expect some canopy storage accounting, which is an aspect that is missed here.*

We have focused on improvements of an interception model rather than multiple related processes. Modeling forest canopy involves several processes, each of which are described with separate models. This includes, unloading, melt and drip (some models), and sublimation. These models use the interception model to dictate how much snow is in the branches at any point in time. If there is still snow in the branches, then it is depleted by the unloading, melt and drip and sublimation models. Thus, canopy storage is dictated by the interplay of each individual model.

We discussed that we present a model for one forest process. To make this more clear we rephrased the explanation (Line 427-431). Furthermore, we made this clear in the method section: "3.2 Subgrid parameterization for forest canopy interception".

[revised manuscript text omitted]

---

## Author Response (AR2)

**We thank the reviewer again for the comments to improve the quality of our manuscript. We have addressed the comments with point-by-point replies to the reviewer (in blue) and revised our manuscript accordingly. Attached is a marked-up version of the manuscript.**

**Reply to reviewer comments:**

*The data and the model equations are better described now. While this clarified things, I still have some rather fundamental concerns.*

• *Snow depth vs SWE: I see the reasoning of the authors, but still would argue that for basically all further uses, one would need SWE. The authors themselves state in the introduction "Accurately modelling the spatial distribution of snow water equivalent in forested regions is thus necessary for climate and water resource modelling over a variety of scales.".*

While we see a benefit of having a reliable SWE interception model (ideally even physical-based and computational efficient) we do not agree that for all further applications one would need SWE interception since snow depth can be converted to SWE by a density model in a snow module (as part of a complex model).

Measuring SWE rather than snow depth of intercepted snow has not been possible (to date) over large scales. Prior studies have been able to accomplish this over the scale of individual trees, by a destructive method which involves cutting a tree and attaching it to a scale in order to derive the weight (and therefore SWE) of intercepted snow.

Given the missing spatial SWE measurements and that converting spatial snow depth to spatial SWE with an empirical density parameterization introduces uncertainties that will be passed on to all model applicants afterwards we do not see a possibility to accomplish a spatial mean SWE model at the moment.

*I am still confused about what the snow height actually refers to. If the intercepted height is 10 cm, does this mean that there are 10 cm snow on the trees or that there is so much snow on the trees that this would be 10 cm if distributed on the ground surface?*

Snow depth interception ($I_{HS}$) describes the snow depth caught by forest canopy. This means that 10 cm intercepted snow leads to 10 cm less snow depth (or height) on the ground. In Section 2 (Data) we now clarify what we mean by "snow depth interception" resp. shorter "snow interception".

*By using heights instead of SWE the model does not necessarily conserve snow masses, which might provide the model with some (unwanted) flexibility. This issue could at least been looked at by estimating densities backwards (assuming conservation of masses).*

We only measured snow depth in forested and open areas, and as such, our model predicts snow height in forested areas based on an open site measurement. The problem of mass conservation is not taken into account in our model as it only deals with the 'loading' phase. What happens after loading is far beyond the scope of this paper, as many other forest processes then come into play (unloading, sublimation, melt and drip). Our interception model thus only provides the input how much snow is in the branches at any point in time.

All snow modules are reliant on a snow density model. We feel this question hits at a long-standing problem of understanding and improving overall model uncertainty from integrated density models, which is not just allocated to interception processes. Improving snow density models is however beyond the scope of this paper.

*Also, when it comes to the conclusions, the use of snow height might cause confusion. What exactly does a statement like "as much as 68 % and on average 43 % of the cumulative snowfall was retained" mean here. Do the % values refer to heights? These might then be quite different from the snow mass (which a not so careful reader of the conclusion might think of).*

When we give percentages how much snow of the cumulative snowfall is retained by forest canopy than this is given as the interception efficiency, which is interception divided by precipitation, i.e. snow depth interception/accumulated snowfall (as indicated in the conclusions). Thus, this measure is independent of units and our conclusions should not cause confusion. We went over the manuscript to check for any ambiguous wording.

• *Model performance: The performance measures need to be better described, with the information in 3.3 it is not possible to reproduce these. For instance, what range was used for normalization (min-max or some percentiles, the latter would probably be more robust).*

We clarified the computation method of the NRMSE's in section 3.3 where we described the performance measures. Indeed, with "normalized by the range of data" we meant the "min-max" for normalization.

• *Model performance: I am also wondering how good the model actually is. The performance measures and the figures look nice, but of course, some of the good-looking performance is rather trivial. Figure 4, for instance, looks good because with larger precip events obviously also the interception increases. The study would be much more convincing if the results of the new model were compared to some baseline estimate. I would recommend using some very basic interception model for comparison to better illustrate the added value of the new approach.*

We agree that it is difficult to assess the performance of an empirical model, especially by only evaluating on the performance of the calibration data set. We therefore gathered additional independent snow interception data sets from different geographical regions and different climate conditions. Figure 4 demonstrates that our empirical model performs similarly well for these two other data sets. Spatial mean interception increases with increasing precipitation but also with increasing $\sigma_z$. Naturally it remains an empirical model and more data sets would be advantageous to validate it in additional regions and climates, but, given the limitation that at the moment there are no more spatial snow depth interception data sets available (due to the inherent difficulties of measuring snow in the canopy over large scales)  which would allow an extended evaluation, our efforts are the best we can do at the moment.

To assess our model performance in the different regions, we give normalized performance measures such as the MPE, MAPE and NRMSE which facilitate performance comparison between different data sets. We obtain similar NRMSE's, MPE's and MAPE's when we apply our model on the different data sets (Table 1). Unfortunately, previous interception models do

not provide relative error measures but give absolute error measures that prevent inter-model comparisons. Since previously presented models were developed for SWE, a direct model comparison of e.g. RMSE's with our interception model developed for snow depth is not possible. Furthermore, previous models were mostly point models and not for spatial mean interception.

Towards a better disclosure of our model performance, we newly manually assessed MPE, MAPE and NRMSE of two previous SWE interception models, namely the stratified $50x50m^2$ model of Moeser et al. (2015) and the point model of Roth et al. (2019). We found overall improved performance measures by our models compared to modeled SWE interception by the two models. We now largely describe this in the discussion and mention it in the conclusion as well as in the abstract.

• _Uncertainties:_ _The authors provide confidence intervals for the different coefficients. This is good, but the more interesting question would be how these translate into uncertainties in the model simulations. For this, all model results should be given with some uncertainty bounds (which could be derived using some MC approach)_

While we agree that such an analysis could be interesting for empirical models developed on large data pools we believe that this analysis would not add more value to our results. We present an empirical model that is based on an extensive intercepted snow depth data set. Based on this spatial data set with about 14'000 individual measurements we derived 60 spatial mean snow depth interception means which forms the data pool (calibration data set) on which we derived our interception model. Validation of our model has been performed using a total of 7 independent spatial mean values. A newly included inter-model comparison with two previous models demonstrates our overall improved model performance. We feel giving uncertainty bounds of modeled interception introduced by the uncertainties of the fit parameters do not provide any extra information to demonstrate our model performance.

• _Applicability of the model elsewhere:_ _Validity for a range of conditions: In the previous round of reviews the issue was raised that the validation sites actually are relatively similar and do not span the potential range of conditions. While the author basically agreed with this in their response, the changes in the text do not fully the potential limitations._

In our last reply we agreed that the novel models should be tested for a broad range of climatic conditions including also extreme climate conditions and at various geographic sites but that we believe that the three sites already cover substantial variability as shown by mean air temperatures and precipitation sums. We therefore pointed out that we believe that the novel models could perform sufficiently well in other climate conditions (though of course extremes have to be investigated). At the moment we do not have more snow interception data sets available (due to the inherent difficulties of measuring snow in the canopy over large scales) that would allow an extended evaluation and we leave this for the future.

We extended this discussion section by additionally comparing error estimates of previous models to those of the presented model here. We believe that the extended section in the discussion improves the overall model applicability discussion.

• _Structure:_ _The authors did not understand the previous comment "Central parts of the methods are described first in the result section.". I am sorry for the confusion and will try to explain this better. The two fundamental equations pop up in the results and it is not clear where they came from. I understand now based on the authors' response that these equations_

*were derived from the Swiss data. Still, this leaves me wondering: was the form of the equation chosen a priori and then parameter values were estimated based on the data or were a number of functions/expressions evaluated? In the first case, I would expect to see some motivation of the expression in the methods, in the latter case I would like to know which range of expressions has been considered and the decision for one or the other has been taken.*

All parameterizations were empirically developed using the Swiss development data set. The existence of varying previously observed functional relationships (base functions) were considered here as well as the correlations between interception and precipitation, $\sigma_z$ and $F_{sky}$ to find an empirical base function as parameterization. During the last revision we largely extended our discussion on our choice of the functional form in the discussion section and added some explanation below the equations in the results section. We now additionally give some details on this in the results section above the equations too.

• *Language: Sorry for repeating this example of ambiguous language:*
*P2L17ff: "In winter as much as 60 % of the cumulative snowfall may be retained in conifer forests"*
*Would snow in another season not be intercepted?*
*".. and as much as 24 % of total annual snowfall may be retained in deciduous forests in the southern Andes"*
*This reads as if 60% of some total snowfall is intercepted in coniferous forests and 24% are intercepted in deciduous forests in the Andes, i.e. 84% are intercepted in total.*
*This is a minor detail and one can guess what the authors mean, but in a scientific paper these things should be formulated as clearly as possible. Here, it should be clarified what the % refers to.*
*As another example: L233ff: "Modeling forest canopy involves several processes such as interception, unloading, melt and drip, and sublimation."*
*Modelling forest canopy would involve rather biological processes, what the author mean is something like 'Modelling the effects of the forst canopy on snow accumulation on the ground ….'*

We rephrased these sentences and went over the manuscript again to check for any ambiguous wording.

[revised manuscript text omitted]

---

## Author Response (AR3)

**We thank the editor for the comment to improve the quality of our manuscript. We have addressed his comment (in blue) and revised our manuscript accordingly. Attached is a marked-up version of the manuscript.**

**Reply to editor comment:**

*While I believe that most of the comments have now been addressed, I think that some of the reviewers' comments have not been correctly interpreted. In particular, the referee pointed towards the need to evaluate the model performance with respect to a baseline interception estimate. Schaefli and Gupta (HYP, 2007) nicely illustrate why such a benchmark could be necessary This is not a matter of additional metrics, but the performance should be evaluated against the performance of a "zero-hypothesis" model. In this case, this could simply be I_HS = c\*P_HS. I think having an additional column in Table 1 which shows the performance of this benchmark model would address this comment of the referee.*

For the last revision we added a model inter-comparison with two promising SWE interception models towards a better disclosure of our model performance. We manually assessed relative error measures (MPE, MAPE and NRMSE) for the previous models, namely the stratified 50x50m$^2$ model of Moeser et al. (2015) and the model of Roth et al. (2019). The stratified model was developed on the most extensive intercepted snow depth data set currently available and the Roth model is the most recently published interception model. We therefore believe that an inter-comparison of relative model performances between these models and our presented models best discloses our model performances. We obtained overall improved performance measures by our snow depth interception models. A description was largely added to the discussion during the last revision.

Following the editor's suggestion we now additionally fitted our development data set to simple baseline models, namely I_HS =cc\*P_HS and $\sigma_{I\_HS}$ =jj\*P_HS to compare our model performances to the performance of the baseline models. Overall, the baseline models perform worse for the development data set as well as for our independent validation data sets. In Table 1 we added two additional lines showing all resulting performance measures for the baseline models. We now introduce the baseline models in the results section and indicate the performances in the validation part of the results section.

To further demonstrate the different performance of the baseline model we give below the corresponding graphs (not shown in the manuscript).

[revised manuscript text omitted]